# Online Reference Tracking For Linear Systems with Unknown Dynamics and Unknown Disturbances

**Nariman Niknejad**                                    *niknejad@msu.edu*
*Department of Mechanical Engineering*
*Michigan State University*
*Michigan, USA*

**Farnaz Adib Yaghmaie**                          *farnaz.adib.yaghmaie@liu.se*
*Faculty of Electrical Engineering*
*Linköping University*
*Linköping, Sweden*

**Hamidreza Modares**                              *modaresh@msu.edu*
*Faculty of of Mechanical Engineering*
*Michigan State University*
*Michigan, USA*

**Reviewed on OpenReview:** *https: // openreview. net/ forum? id= pfbVayaUMc*

## Abstract

This paper presents an online learning mechanism to address the challenge of state tracking for unknown linear systems under general adversarial disturbances. The reference trajectory is assumed to be generated by unknown exosystem dynamics, which relaxes the common assumption of known dynamics for exosystems. Learning a tracking control policy for unknown systems with unknown exosystem dynamics under general disturbances is challenging and surprisingly unsettled. To face this challenge, the presented online learning algorithm has two stages: In the first stage, an algorithm identifies the dynamics of the uncertain system, and in the second stage, an online parametrized memory-augmented controller accounts for the identification error, unknown exosystem dynamics as well as disturbances. The controller's parameters are learned to optimize a convex cost function, which is not necessarily quadratic, and learning the control parameters is formulated as an online convex optimization problem. This approach uses the memory of previous disturbances and reference values to capture their effects on performance over time. Besides, it implicitly learns the dynamics of the exosystems. The algorithm enables online tuning of controller parameters to achieve state tracking and disturbance rejection. It is shown that the algorithm achieves a policy regret of $\mathcal{O}(T^{2/3})$. In the simulation results, the performance of the presented tracking algorithm is compared with the certainty equivalent $H_\infty$-control and linear quadratic regulator.

## 1 Introduction

Reference tracking is a fundamental problem in control theory (Isidori, 1985; Huang, 2004; Dixon et al., 2004; Vamvoudakis et al., 2017) and it has many applications (Zare et al., 2022), where the goal is to design a control policy that steers the closed-loop system towards a reference trajectory or set-point. Significant progress has been made toward developing reference tracking controllers for both linear and nonlinear systems. Most of the existing results, however, have made all or some of the following assumptions: 1) the system dynamics are known; 2) the exosystem dynamics generating the reference trajectories are known; 3) the system is deterministic and might be under bounded energy disturbances or the system is stochastic and

under independent and identically distributed (i.i.d.) noise, mostly Gaussian noise; and 4) only asymptotic tracking is of concern, and no optimality of performance is concerned.

Several learning-based or adaptive controllers have been presented to deal with epistemic uncertainties (i.e., uncertainties that can be reduced by collecting data) in system dynamics and exosystem dynamics (i.e., to relax numbers 1 and 2 of the above-mentioned assumptions). Traditional adaptive controllers do not provide performance guarantees, as they only optimize an instantaneous cost function (Lewis, 1986; Sutton & Barto, 2018; Bertsekas, 2019; Zhang & Lewis, 2012). In contrast, reinforcement learning (RL) (Yao & Yao, 2022; Wang et al., 2022b; Chen et al., 2019; Gao & Jiang, 2022; Deng et al., 2021; Mei et al., 2022; Modares et al., 2016; Rabiee & Safari, 2023) leads to learning adaptive optimal control policies by optimizing a long-horizon cost function using collected data. Despite this advantage, existing RL-based control solutions for continuous state-actions are limited to deterministic systems with no disturbances (Li & Wu, 2020; Hao et al., 2021) or bounded disturbances (Li et al., 2020; Mohammadi et al., 2021) and stochastic systems with Gaussian noise (Cheng et al., 2019). For the case of bounded disturbances, RL algorithms typically reformulate the $H_2$-optimal control design into a robust min-max or $H_\infty$-optimal control problem (Khalil, 2002; Modares et al., 2015), which can be overly conservative. To this end, some works proposed combining controllers such as resonant control with $H_\infty$ for a better performance (Dadkhah & Moheimani, 2023). For stochastic systems, existing RL algorithms are either based on policy gradient under which the controller is parametrized and its parameters are learned through the gradient descent method or based on policy iteration method under which the policies are iteratively evaluated until convergence. The former requires approximating the expected gradient of the cost function, and the latter requires approximating the expected cost itself. These results are typically limited to systems with Gaussian noise. However, in practical control systems, adversaries aiming to degrade the control performance can act as adversarial disturbances that are unpredictable and do not follow any distribution.

In this paper, the main focus is on constructing tracking controllers for linear systems when the underlying system dynamics and exosystem dynamics are unknown, and disturbances are adversarial. Adversarial disturbances can take any arbitrary form and are not restricted to those that are bounded with a known bound or follow a probability distribution. A control policy in the form of (disturbance, reference)-action is developed that leverages a fixed-size history of disturbances and reference values in its actions. The presented online learning algorithm extends the results of Yaghmaie & Modares (2023); Hazan et al. (2020) to systems with unknown dynamics and has two stages: In the first stage, a system identifier algorithm estimates the unknown dynamics and uncertainty, and in the second stage, an online parameterized memory-augmented controller is learned to optimize a convex cost function while accounting for the identification error, unknown exosystem dynamics, and adversarial disturbances. Yaghmaie & Modares (2023) provides a concise parameterization of the control policy resulting in $\mathcal{O}(\sqrt{T})$ regret bound benchmarking against the best linear control policy when the system dynamics is known. In this extension to their work, our algorithm achieves a regret bound of $\mathcal{O}(T^{2/3})$ for linear systems with *unknown* dynamics. Besides the theoretical guarantees, in the simulation results, we compare the performance of the presented algorithm with a couple of relevant solutions including the $H_\infty$ and Linear Quadratic Regulator (LQR) control to highlight its superior performance.

## 2 Related works

In this section, the related works to the problem of optimal tracking are summarized. These works are focused on scenarios with an induced disturbance on the states and also where the dynamics of the system are unknown.

**System identification:** Linear dynamic system identification is the process of determining the mathematical model that describes the input-output behavior of a linear dynamic system and it has been studied in Ljung (1998). The least-squares method and its variants, such as total least squares and recursive least squares, are widely used to identify the system's parameters (Tatari et al., 2021; Faradonbeh et al., 2017; Sarkar et al., 2019). System identification using machine learning techniques, such as artificial neural networks and support vector machines, has also gained popularity in recent years (Nagumo & Noda, 1967; Weber et al., 2019; Chiuso & Pillonetto, 2019; Mehrzad et al., 2023). The choice of method depends on

the system's characteristics, the available data, and the required accuracy of the identified model. A properly identified model can facilitate the design of robust controllers and the prediction of system behavior under different operating conditions. In situations where there is an adversarial disturbance, the use of the least-squares method may produce unreliable estimates. Thus, this paper exploits the method introduced by (Theorem 19 in (Hazan et al., 2020)).

**Output regulation theory:** The output regulation theory, as introduced by Isidori and Huang in their works (Isidori, 1985; Huang, 2004), has been widely utilized in the design of model-free reinforcement learning (RL) algorithms for solving optimal tracking problems, as well as in attenuating the effects of disturbances (Gao et al., 2017; Chen et al., 2022; Jiang et al., 2020b; Chen et al., 2019; Jiang et al., 2020a; Gao & Jiang, 2016; 2015). However, a limitation of RL and adaptive dynamic programming (ADP) approaches based on the output regulation theory is that they assume the disturbance is generated by a dynamical system, which is not always the case in many real-world applications. This constraint restricts the applicability of the output regulation theory in practical scenarios. Additionally, ADP methods typically optimize risk-neutral (expected) or risk-aware measures of the cost function under the assumption of i.i.d and Gaussian noise. This assumption is made because either the value function is learned directly based on collected data to estimate expected or risk-aware accumulated rewards in policy interaction or value iteration methods, or the expected or risk-aware cost function or its derivative with respect to control parameters is learned from data in policy gradient methods. For general disturbances, usually a robust control approach is utilized which is discussed below.

**Robust control design:** To handle general disturbances with limited energy, the $H_\infty$-control theory is often employed to guarantee an $\mathcal{L}_2$-gain performance bound (Doyle, 1995; Khalil, 2002; Modares et al., 2015). That said, the $H_\infty$-approach is known to be overly conservative, as the resulting robust controller is designed to hedge against the worst-case disturbance sequence, which is rarely encountered in reality. To that end, some works proposed online compensation for unknown stochastic disturbances for motion planning and control (Faust et al., 2015).

**Gaussian disturbance:** The Linear Quadratic Regulator (LQR) can be used to design an optimal controller for linear systems subject to Gaussian process disturbance (noise) (Bertsekas, 2012). It is also the optimal controller for noise-free linear systems. However, in many practical control systems, the disturbance does not follow a Gaussian distribution or the cost function is not quadratic. The provided guarantees so far in the literature are for the Gaussian case.

## 3 Optimal Reference Tracking Problem

*Notations and preliminaries:* Let $I$ represent an identity matrix with the appropriate size. Let **1** and **0** denote matrices with appropriate sizes consisting of all ones and all zeros, respectively. The gradient of a function $f(x)$ with respect to $x$ is denoted by $\nabla_x f$. The $\mathcal{L}_2$-norm of $x$ is denoted by $\|x\|_{\mathcal{L}_2} = (\sum_{k=0}^{+\infty} \|x_k\|^2)^{\frac{1}{2}}$ where $\|x_k\|$ is the instantaneous Euclidean norm of the vector $x_k$. For matrix $A$, the spectral norm is denoted by $\|A\|$, and the Frobenius norm is denoted by $\|A\|_F$. Let $\mathbb{I}_E$ be an indicator function on set $E$. For a time-dependent variable $x_k$, the notation $x_{i:j}$, $j \geq i$ is defined as $x_{i:j} = \{x_i, x_{i+1}, .., x_j\}$. The notation $\mathcal{O}()$ is leveraged throughout the paper to express the regret upper bound as a function of $T$.

### 3.1 Tracking Problem

Consider the following linear dynamical system

$$x_{k+1} = Ax_k + Bu_k + w_k, \tag{1}$$

where the variables $x_k \in \mathbb{R}^n$ and $u_k \in \mathbb{R}^m$ represent the state and control input of the system, respectively. In equation 1, $w_k \in \mathbb{R}^n$ denotes the adversarial (unknown and arbitrary) disturbance. Only a bound, which also does not need to be known a priori, is assumed on the disturbance for theoretical reasons. It can be assumed that $x_0 = \mathbf{0}$ without loss of generality and incorporate the initial condition into $w_0$.

The objective of this paper is to choose the input variable $u_k$ in such a way that the state of the system $x_k$ follows an arbitrary reference signal $r_k$ that is not known beforehand. This reference signal is only revealed

sequentially after the control input has been applied.

$$z_{k+1} = Sz_k,$$
$$r_k = Fz_k, \tag{2}$$

where $z_k \in \mathbb{R}^p$, $r_k \in \mathbb{R}^n$ represent the state and the output of the reference generator.

In the sequel, a list of a few definitions and results are brought that are related to equation 1-equation 2 and the tracking problem.

**Definition 1** *(Agarwal et al., 2019) Consider*

$$x_{k+1} = Ax_k + Bu_k,$$

*and $\gamma \in [0,1)$, $\kappa > 1$. A linear controller $K$ is $(\kappa, \gamma)$-stable if $\|K\| \leq \kappa$ and $\|\tilde{A}_K^t\|_2 \leq \kappa^2(1-\gamma)^t \ \forall t \geq 0$ where $\tilde{A}_K = A + BK$. Equivalently, a linear controller $K$ is also $(\kappa, \gamma)$-stable if there exist a decomposition of $\tilde{A}_K = QLQ^{-1}$, such that $\|L\| \leq (1-\gamma)$, and $\|A\|, \|B\|, \|Q\|, \|Q^{-1}\|, \|K\| \leq \kappa$.*

**Definition 2** *(Strong Controllability)(Definition 7 in (Hazan et al., 2020)) A linear dynamical system $(A, B)$ is said to have controllability index $\lambda$ if the matrix $G_\lambda$ is full-row rank, and*

$$G_\lambda = [B, AB, A^2B...A^{\lambda-1}B],$$

*where $G_\lambda$ is defined as the matrix associated with $(A, B)$ for $\lambda \geq 1$. In addition, such a system is also defined $(\lambda, \kappa)$ strongly controllable if $\|(G_\lambda G_\lambda^T)^{-1}\| \leq \kappa$.*

In a controllable system, the controllability index $\lambda$ has an upper bound of the number of states in the system. It is worth noting that, due to the Cayley-Hamilton theorem, the controllability index of a controllable system is never greater than the dimension of the state space. Assuming that the system $(A + BK, B)$ is $(\lambda, \kappa)$ strongly controllable, similar to the concept of stability, a measurable counterpart of controllability is initially presented by (Cohen et al., 2018).

**Lemma 1** *(Maintaining Stability) (Lemma 15 in (Hazan et al., 2020)) Consider an identified dynamical system with $(\hat{A}, \hat{B})$. Assume that the original system is $(\kappa, \gamma)$-strongly stable. It can be shown that control gain $K$ is $(\kappa + \epsilon_{A,B}, \gamma - 2\kappa^3\epsilon_{A,B})$- strongly stable for $(\hat{A}, \hat{B})$, as long as $\|A - \hat{A}\|, \|B - \hat{B}\| \leq \epsilon_{A,B}$. For the system with estimated matrices, one has $\|\hat{A}\|, \|\hat{B}\| \leq \kappa + \epsilon_{A,B}$. Assuming $\|Q\|, \|Q^{-1}\|, \|K\|, \|A\|, \|B\| \leq \kappa$, we define $\hat{A} + \hat{B}K = Q\hat{L}Q^{-1}$ where one can show $\|\hat{L}\| \leq 1 - \gamma + 2\kappa^3\epsilon_{A,B}$, where $\hat{L} := L + Q^{-1}((\hat{A} - A) - (\hat{B} - B)K)$. Moreover, the $Q$ matrix coincide in both the actual system $(A, B)$ and the one with estimated system matrices $(\hat{A}, \hat{B})$.*

Theorem 1 is typically leveraged to present a fundamental discovery outlining the necessary and sufficient condition for the existence of a linear feedback strategy that can solve the state tracking problem. Specifically, the problem concerns ensuring that $x_k \rightarrow r_k$ in the absence of disturbances. In this context, the term "linear feedback policy" denotes a particular approach utilized to address the problem.

**Theorem 1** *(Isidori, 1985) Consider the dynamical system in equation 1 and the reference signal in equation 2. Assume that $w_k \equiv \boldsymbol{0}$, $(A, B)$ is stabilizable. Assume that the learner has previous knowledge on $K_{fb}$ such that $A + BK_{fb}$ is strongly stable. Then, the controller*

$$u_k^{lin}(K_f) = K_{fb}x_k + K_{ff}z_k. \tag{3}$$

*solves the classical state tracking problem $x_k \rightarrow r_k$, if and only if there exist matrices $\Pi \in \mathbb{R}^{n \times p}$ and $\Gamma \in \mathbb{R}^{m \times p}$ such that*

$$\Pi S = A\Pi + B\Gamma, \ \Pi - F = \boldsymbol{0} \tag{4}$$

*and $K_{ff} = \Gamma - K_{fb}\Pi$.*

### 3.2 Performance index

The main goal of this paper is to design a control policy $\pi : (x_{1:k}, w_{1:k-1}, r_{1:k}) \to u_k$ that optimizes an average cost function, reflecting the designer's intentions. The total cost linked to a given policy $\pi$ is determined as follows

$$J_T(\pi) = \sum_{k=1}^{T} c_k(x_k, u_k), \tag{5}$$

where $c_k$ is the rolling cost. Also, the average cost of a policy $\pi$ is defined as the below equation

$$\bar{J}_T(\pi) = \frac{1}{T} \sum_{k=1}^{T} c_k(x_k, u_k). \tag{6}$$

### 3.3 The presented policy

The conventional linear controller in the form of equation 3 aims to mitigate the impact of adversarial or arbitrary disturbances on the cost function by determining gains $K_{fb}$ and $K_{ff}$ using $H_\infty$-control design. However, this approach is overly conservative and encounters difficulties in online control design due to the non-convexity of the cost function $c_k(x_k, u_k)$ with respect to $K_{fb}$ and $K_{ff}$. To tackle this issue, we work with the class of memory augmented policies which is capable of handling adversarial disturbances and unknown dynamics (Yaghmaie & Modares, 2023; Agarwal et al., 2019).

**Definition 3** *A Memory-augmented Control Policy is denoted by $\pi(K, M, P)$*

$$u_k^\pi(K, M, P) = Kx_k + \sum_{t=1}^{m_w} M^{[t-1]} w_{k-t} + \sum_{s=0}^{m_r-1} P^{[s]} r_{k-s}, \tag{7}$$

*where $K \in \mathcal{K}$ is a fixed matrix and $Y = [M, P] = [M^{[0]}, ..., M^{[m_w-1]}, [P^{[0]}, ..., P^{[m_r-1]}] \in \mathcal{Y}$ are parameters to be learned. The domains $\mathcal{K}$, $\mathcal{Y}$ are defined as*

$$\mathcal{K} = \{K : A + BK \text{ is } (\kappa, \gamma) - stable\}, \tag{8}$$
$$\mathcal{Y} = \{Y = [M^{[0]}, ..., M^{[m_w-1]}, P^{[0]}, ..., P^{[m_r-1]}] | \|M^{[t]}\|, \|P^{[t]}\| \leq \kappa_b \kappa^3 (1-\gamma)^t\}.$$

Since the policy parameters are learned, which are changing over time, $M_k = [M_k^{[0]}, ..., M_k^{[m_w-1]}]$ and $P_k = [P_k^{[0]}, ..., P_k^{[m_r-1]}]$ are representing as the policy parameters at step $k$.

Observer that the class of memory-augmented policies is more general than the class of linear controller policies. Indeed, a linear control policy is a special case of the memory-augmented policy.

Let $x_k^\pi$ be the state attained upon execution of the policy $\pi(K, M_{0:k-1}, P_{0:k-1})$ that generates the control input in equation 7 at time $k$. One can show that the state attained upon execution of a memory-augmented control policy is linear in $M$. This is established in the next lemma. Consequently, this implies linearity of the memory-augmented control policy in $M$. Since the cost function $c_k(x_k, u_k)$ is convex in $x_k, u_k$ and $x_k^\pi$, $u_k^\pi$ are linear in $M$, one can conclude that $c_k$ is convex in $M$. Let $\tilde{A}_K = A + BK$ and define

$$\Psi_{k,y}^{K,h}(M_{k-h-1:k-1}) := \tilde{A}_K^y \mathbb{I}_{y \leq h-1} + \sum_{j=0}^{h-1} \tilde{A}_K^j B M_{k-j-1}^{[y-j-1]} \mathbb{I}_{1 \leq y-j \leq m_w}, \tag{9}$$

$$\psi_{k,z}^{K,h}(P_{k-h-1:k-1}) := \sum_{j=0}^{h-1} \tilde{A}_K^j B P_{k-j-1}^{[z-j-1]} \mathbb{I}_{1 \leq z-j \leq m_r}. \tag{10}$$

**Lemma 2 ((Yaghmaie & Modares, 2023))** *Let $x_k^\pi$ be the state attained upon execution of the policy $\pi(K, M_{0:k-1}, P_{0:k-1})$ that generates the control input in equation 7 at time k. Then*

$$
\begin{aligned}
x_k^\pi = x_k^K(M_{0:k-1}, P_{0:k-1}) = &\tilde{A}_K^h x_{k-h}^\pi + \sum_{y=0}^{m_w+h-1} \Psi_{k,y}^{K,h}(M_{k-h-1:k-1}) w_{k-y-1} \\
&+ \sum_{z=0}^{m_r+h-1} \psi_{k,z}^{K,h}(P_{k-h-1:k-1}) r_{k-z}.
\end{aligned}
\tag{11}
$$

*or equivalently*

$$
x_k^\pi = x_k^K(M_{0:k-1}, P_{0:k-1}) = \sum_{y=0}^{k-1} \Psi_{k,y}^{K,k}(M_{0:k-1}) w_{k-y-1} + \sum_{z=0}^{k-1} \psi_{k,z}^{K,k}(P_{0:k-1}) r_{k-z}.
\tag{12}
$$

### 3.4 Assumptions and the optimal tracking problem

In this subsection, a list of the assumptions is brought to be used throughout the paper and define the optimal tracking problem in the presence of adversarial disturbances.

**Assumption 1 (dynamical system)** *The pair $(A, B)$ is unknown but stabilizable. The actual system matrices $(A, B)$ and the identified system dynamics matrices $(\hat{A}, \hat{B})$ are bounded, i.e., $\|A\|, \|B\| \leq \kappa$ and $\|\hat{A}\|, \|\hat{B}\|, \leq \kappa + \epsilon_{A,B}$, where $\epsilon_{A,B}$ denotes the distance between the actual and identified system matrices.*

**Assumption 2 (disturbance)** *The disturbance sequence $w_k$ is bounded, i.e., $\|w_k\| \leq \kappa_w$ for some $\kappa_w > 0$. Moreover, the disturbance $w_k$ does not depend on the control input $u_k$.*

**Assumption 3 (reference signal)** *The dynamics of the reference signal generator are unknown but detectable. The state of the reference signal $z_k$ is not measurable but the output $r_k$ is measurable and $r_k$, $z_k$ are bounded, i.e., $\|r_k\| \leq \kappa_r$ and $\|z_k\| \leq \kappa_z$.*

**Assumption 4 (Known Linear Controller)** *A control gain $K$ in equation 7 that makes the unknown system $(A, B)$, $(\kappa, \gamma) - stable$ is available to the learner. In other words, the set in equation 8 is known.*

**Remark 1** *Partial knowledge of the system model can be typically extracted for many practical systems using the physical information available, which can be leveraged to design robust controllers satisfying Assumption 4. That is, even though no knowledge of the system models is used during learning, partial knowledge of the system models (e.g., the matrices $A$ and $B$ belonging to sets of possible system models) is required to find the starting control policy. This is a standard assumption in most control systems since, without a stabilizing control policy to start with, the system's state can quickly become large and useless to learn from, and the system can also fail before any learning occurs.*

Assumption 3 stipulates that the reference signal must be bounded, as an unbounded reference signal may lead to an unbounded average cost. This assumption is commonly employed in analyzing average cost, as observed in works like Abbasi-Yadkori et al. (2014) and Adib Yaghmaie et al. (2019). Nevertheless, the issue of tracking unbounded reference signals can be tackled by exploring discounted cost settings, where an appropriate discounting factor can ensure the boundedness of the discounted cost. This approach has been demonstrated in other studies, including Kiumarsi et al. (2014).

**Assumption 5 (cost function)** *The cost $c_k(x_k, u_k)$ is convex in $x_k$, $u_k$. Moreover, when $\|x\|, \|u\| \leq D$, it holds that $|c_k(x_k, u_k)| \leq \beta D^2$ and $\|\nabla_x c_k(x, u)\|, \|\nabla_u c_k(x, u)\| \leq G_c D$ for some $\beta > 0$ and $G_c > 0$.*

Assumption 5 broadens the scope of applicable cost functions beyond quadratic forms, thereby enhancing the inclusiveness of the assumption.

**Problem 1 (Optimal Tracking Against Adversarial Disturbances with Unknown Dynamics)**
*Consider the dynamical system in equation 1, the reference generator in equation 1-equation 2 and the cost function in equation 5. Let Assumptions 1-5 hold. Design a policy in the form of equation 7*

$$u_k^\pi(K, M, P) = Kx_k + \sum_{t=1}^{m_w} M^{[t-1]}w_{k-t} + \sum_{s=0}^{m_r-1} P^{[s]}r_{k-s},$$

*from the class of memory-augmented policies in Definition 3 to optimize the total cost in equation 5*

$$J_T(\pi) = \sum_{k=1}^{T} c_k(x_k, u_k^\pi).$$

## 4 Memory-augmented online state-tracking algorithm

We propose Algorithm 1 to solve Problem 1. The algorithm uses the concept of truncated state and cost which will be defined in the sequel.

### 4.1 Truncated state, input and cost

Similar to (Yaghmaie & Modares, 2023), we limit everything to a fixed memory length of $H$. Let $\tilde{x}_k^\pi, \tilde{u}_k^\pi, : f_k$ represent the truncated state, input, and cost if the system had started at $\tilde{x}_{k-H}^\pi = \mathbf{0}$. The expressions for $\tilde{x}_k^\pi$, $\tilde{u}_k^\pi$ are

$$\tilde{x}_k^\pi(M_{k-H-1:k-1}, P_{k-H-1:k-1}) = \tag{13}$$
$$\sum_{y=0}^{m_w+H-1} \Psi_{k,y}^{K,H}(M_{k-H-1:k-1})w_{k-y-1} + \sum_{z=0}^{m_r+H-1} \psi_{k,z}^{K,H}(P_{k-H-1:k-1})r_{k-z},$$

$$\tilde{u}_k^\pi(M_{k-H-1:k}, P_{k-H-1:k}) = \tag{14}$$
$$K\tilde{x}_k^K(M_{k-H-1:k-1}, P_{k-H-1:k-1}) + \sum_{t=1}^{m_w} M_k^{[t-1]}w_{k-t} + \sum_{s=0}^{m_r-1} P_k^{[s]}r_{k-s},$$

and the truncated cost $f_k$ reads

$$f_k(M_{k-H-1}, ..., M_{k-1}, P_{k-H-1}, ..., P_{k-1})$$
$$= c_k(\tilde{x}_k^\pi(M_{k-H-1:k-1}, P_{k-H-1:k-1}) - r_k, \tilde{u}_k^\pi(M_{k-H-1:k}, P_{k-H-1:k})). \tag{15}$$

In Appendix B, some theoretical results are provided regarding the memory-augmented controllers and the associated states and costs which are essential in obtaining the main result in Theorem 4.

### 4.2 The overall online learning algorithm (Algorithm 1)

Algorithm 1 involves two stages. In the first stage, the system dynamics are identified, then, in the second stage, the online controller is learned. This approach is commonly known as the explore-then-commit pipeline and will be explained in the sequel.

The algorithm starts in **Line 1** with the selection of a stabilizing controller gain $K$, as well as other necessary parameters.

**System identification (Algorithm SysId):** Following the initiation of the algorithm, the identification stage begins by executing Algorithm SysId. As detailed in Algorithm SysId, the controller in equation 18 is used to collect $T_0$ samples. In this algorithm, it is assumed that the learner has access to a stabilizing

control gain $K$. System identification using binary inputs with values of -1 and 1 (also called bipolar inputs) is a method to analyze and model the behavior of a system. By systematically applying binary input sequences consisting of -1 and 1, the system's response is measured and recorded in **Line 3**.In **Lines 4,5,6**, dummy variables $G$ and $Q$ are computed. Then, in **Line 7**, a deterministic-equivalent matrix pair $(\hat{A}, \hat{B})$ is identified through an iterative procedure that determines matrices of the form $(A + BK)^i B$, followed by solving a linear system of equations to recover the original matrix $A$. Note that during this stage, the trajectory to be followed is disregarded. The estimated dynamics $(\hat{A}, \hat{B})$ is fed to the presented robust tracking algorithm. This identification procedure is inspired by (Hazan et al., 2020).

**Remark 2** *(Ljung, 1995) Generating a stochastic binary signal often involves the introduction of white Gaussian noise, which is subsequently filtered using a carefully selected linear filter. The resulting signal's sign is then extracted, conforming it to a desired binary level. A signal is deemed favorable for the identification of linear systems when it exhibits a small crest factor. The crest factor, with a minimum value of 1, is achieved when employing a binary waveform. However, binary signals prove less useful for nonlinear system identification due to their limited information content and lack of varied excitation levels (Novak et al., 2009).*

**Robust tracking (Algorithm RobTrack):** Upon identification of the system, the domain set $\mathcal{Y}$ is initialized and a loop starts. In **Line 7**, the reference signal $r_k$ is recorded. $u_k^\pi$ in equation 7 is calculated and applied to the system. Next, in **Line 8**, the next state $x_{k+1}$ is observed, and the disturbance $\hat{w}_k$ is estimated by equation 16. Selection of $\hat{w}_k$ according to equation 16 ensures that the state, action, and cost produced by Algorithm 1 coincide with those of the actual system. In **Line 9**, the algorithm suffers the cost $c_k(e_k, u_k)$. Then **Line 10**, the truncated state and inputs are computed from equation 13-equation 14 using the latest values of $M, P$, and the truncated cost $f_k(M^{[0]}, ..., M^{[m_w-1]}, P^{[0]}, ..., P^{[m_r-1]})$ is calculated from equation 15. In **Line 11**, the weights $M$, $P$ are adjusted with projected gradient descent on the truncated cost $f_k(M^{[0]}, ..., M^{[m_w-1]}, P^{[0]}, ..., P^{[m_r-1]})$ based on equation 17.

It should be noted that during each iteration $k$ of the algorithm outlined in Algorithm 1, the values of $\hat{w}_{T_0+1:k-1}$ and $r_{T_0+1:k}$ are already known and accessible. Additionally, for all $k < T_0$, $\hat{w}_k$ and $r_k$ are defined to be equal to the zero vector. Thus, it is possible to compute the expressions in equation 13-equation 15 for any given iteration of the algorithm. The projection operator $\Pi_M$ and $\Pi_P$ are matrix projection operators with $\mathcal{L}_2$ norm of $\kappa_b \kappa^3 (1 - \gamma)$.

The properties related to Algorithms SysId and 1 are given in Appendices C-D.

## 4.3 Regret Analysis

The standard measure for online control based on the gradient descent is the policy regret (Agarwal et al., 2019), which is defined here as the difference between cumulative cost of the designed parameterized control policy $\pi$ learned by Algorithm 1 and that of the optimal linear control policy in the form of equation 3.

**Definition 4** *Consider the system in equation 1. Let the control policy be designed to generate the control action $u_k$ in equation 7 at time $k$. Let Algorithm 1 be used to update the parameters of $u_k$. Then, its regret is defined as*

$$Regret = \sum_{k=1}^{T} c_k(x_k, u_k) - \min_{K_f \in \mathcal{K}} J_T(K_f),$$

*where $J_T(K_f)$ is the total cost in equation 5 of the linear feedback controller in equation 3.*

The regret compares the performance of Algorithm 1 generating controllers from the class of feasible memory-augmented control policies with the best linear control policy in hindsight.

---

**Algorithm 1** Online state tracking algorithm

---

1: **Initialize:** Set a stabilizing controller gain $K$, perturbation horizon $m_w$, reference horizon $m_w$, rounds of system identification $T_0$, number of iterations after system identification $T$, and horizon of identification $\lambda$.

2: **Stage 1: System Identification**

$$(\hat{A}, \hat{B}) = \text{SysId}(T_0, \lambda).$$

3: **Stage 2: Robust Tracking (Algorithm RobTrack)**

4: Initialize $\mathcal{Y} = \{Y = [M^{[0]}, ..., M^{[m_w-1]}, \ P^{[0]}, ..., P^{[m_r-1]}] | \|M^{[t]}\|, \|P^{[t]}\| \le \kappa_b \kappa^3 (1-\gamma)^t \}$ .

5: Set $\hat{w}_k = 0$ for all $k \le T_0$ and $\hat{w}_k = x_{T_0}$.

6: **for** $k = T_0 + 1, .., T_0 + T$ **do**

7:     Record $r_k$ and execute

$$u_k^\pi(K, M, P) = Kx_k + \sum_{t=1}^{m_w} M^{[t-1]}\hat{w}_{k-t} + \sum_{s=0}^{m_r-1} P^{[s]}r_{k-s}$$

8:     Observe $x_{k+1}$ and record an estimate

$$\hat{w}_k = x_{k+1} - \hat{A}x_k - \hat{B}u_k. \tag{16}$$

9:     Suffer $c_k(e_k, u_k)$.

10:    Compute $f_k(M^{[0]}, ..., M^{[m_w-1]}, P^{[0]}, ..., P^{[m_r-1]})$ in equation 13–equation 15 for $\hat{A}$, $\hat{B}$, and $\hat{w}$.

11:    Update $M$, $P$.

$$M = \Pi_M(M - \eta \nabla_M f_k(M^{[0]}, ..., M^{[m_w-1]}, P^{[0]}, ..., P^{[m_r-1]})),$$
$$P = \Pi_P(P - \eta \nabla_P f_k(M^{[0]}, ..., M^{[m_w-1]}, P^{[0]}, ..., P^{[m_r-1]})). \tag{17}$$

---

---

**Algorithm SysId** System identification by inducing random inputs

---

1: **Inputs:** $T_0$, $\lambda$.

2: **for** $k = 1, ..., T_0$ **do**

3:     Induce the control

$$u_k = Kx_k + \eta_k, \eta_k \sim_{i.i.d.} \{\pm 1\}^m. \tag{18}$$

4:     Observe and record the resulting state $x_k$.

5: Calculate $Q_j = \frac{1}{T_0 - \lambda} \sum_{k=0}^{T_0 - \lambda - 1} x_{k+j+1} \eta_k^T, \ \forall j \in \{\lambda\}$.

6: Form $G_0 = (Q_0, ..., Q_{\lambda-1})$, $G_1 = (Q_1, ..., Q_\lambda)$.

7: **Outputs:** $\hat{A}$ and $\hat{B}$

$$\hat{B} = Q_0, \ \hat{A}' = G_1 G_0^T (G_1 G_0)^{-1}, \ \hat{A} = \hat{A}' - \hat{B}K.$$

---

In the sequel, we give the regret analysis of Algorithm 1. The main technical difficulty in the regret analysis lies in combining the system identification and the online control approach where both the estimated and true dynamics are present. Another aspect is to find the right balance between the identification and control horizons to guarantee a sublinear regret bound.

**Theorem 2** *Suppose Algorithm 1 is executed under Assumptions 1-5. Let $H = m_w = m_r$. Select the learning rate $\eta$ and the memory size $H$ to satisfy $\eta = \mathcal{O}(\frac{1}{G_c \kappa_w \sqrt{T}})$, $H = \mathcal{O}(log \frac{\kappa^2 T}{\gamma})$, and $T_0 = T^{2/3}$. Then,*

$$Regret = \mathcal{O}(T^{2/3}).$$

*Proof:* Let $K^* = \arg\min_{K \in \mathcal{K}} \sum_{k=1}^{T} c_k(x_k, u_k)$. We decompose the regret as

$$
\begin{aligned}
\text{Regret} &= \sum_{k=1}^{T} (c_k(x_k, u_k) - \min_{K \in \mathcal{K}} c_k(x_k, u_k)) \\
&= \sum_{k=1}^{T_0} (c_k(x_k, u_k) - \min_{K \in \mathcal{K}} c_k(x_k, u_k)) + \sum_{k=T_0+1}^{T} (c_k(x_k, u_k) - \min_{K \in \mathcal{K}} c_k(x_k, u_k)) \\
&\leq \sum_{k=1}^{T_0} (c_k(x_k, u_k) - \min_{K \in \mathcal{K}} c_k(x_k, u_k)) + \sum_{k=1}^{T} (c_k(x_k, u_k) - \min_{K \in \mathcal{K}} c_k(x_k, u_k)) \\
&= \underbrace{\sum_{k=1}^{T_0} (c_k(x_k, u_k) - \min_{K \in \mathcal{K}} c_k(x_k, u_k))}_{J_0} + \underbrace{J(\mathcal{A}|\hat{A}, \hat{B}, \{\hat{w}\}, \{r\}) - J(K^*|\hat{A}, \hat{B}, \{\hat{w}\}, \{r\})}_{R_1} \\
&\quad + \underbrace{J(K^*|\hat{A}, \hat{B}, \{\hat{w}\}, \{r\}) - J(K^*|A, B, \{w\}, \{r\})}_{R_2}.
\end{aligned}
$$

The term $J_0$ contains the regret for the system identification stage in Algorithm SysId. The regret analysis is given in Lemma 8 where we show that $R_1 = \mathcal{O}(T_0)$. The term $R_1$ compares the total cost by our algorithm using the estimated dynamics with that of the best linear controller using the estimated dynamics. In Theorem 4, we prove that $R_2 = \mathcal{O}(\sqrt{T})$. The term $R_2$ compares the total cost by the best linear controller using the estimated dynamics with that of using the original dynamics. In Lemma 11, we show that $R_2 = \mathcal{O}(T T_0^{-1/2})$. Selecting $T_0 = T^{2/3}$, the regret is concluded.

## 5 Simulation results

In this section, the simulation results are given.

### 5.1 The dynamical system, reference, and cost function

Consider the following tracking problem where the dynamics of the system is considered as

$$x_{k+1} = \begin{bmatrix} 1 & 1 \\ 0 & 1 \end{bmatrix} x_k + \begin{bmatrix} 1 & 0 \\ 0 & 1 \end{bmatrix} u_k + w_k, \tag{19}$$

and the reference signal is generated by

$$
\begin{aligned}
z_{k+1} &= \begin{bmatrix} 0 & 1 & 0 \\ -1 & 1.5 & 0 \\ 0 & 0 & 1 \end{bmatrix} z_k, \ z_0 = [1, -2, 0.5]^T, \\
r_k &= \begin{bmatrix} 1 & 0 & 0 \\ 0 & 0 & 1 \end{bmatrix} z_k.
\end{aligned} \tag{20}
$$

where

$$x_k = \begin{bmatrix} x_{1k} \\ x_{2k} \end{bmatrix}, \ w_k = \begin{bmatrix} w_{1k} \\ w_{2k} \end{bmatrix}, \ r_k = \begin{bmatrix} r_{1k} \\ r_{2k} \end{bmatrix}, \ e_k = \begin{bmatrix} e_{1k} \\ e_{2k} \end{bmatrix} = \begin{bmatrix} x_{1k} - r_{1k} \\ x_{2k} - r_{2k} \end{bmatrix}.$$

A quadratic cost with $Q = 20I_2$, $R = I_2$ is considered; that is

$$c_k = e_k^T Q e_k + u_k^T R u_k.$$

Note that the presented algorithm is designed to handle any convex cost function, but a quadratic cost is chosen for comparison with classical control approaches such as Linear Quadratic Regulator (LQR) and $H_\infty$-controllers.

### 5.2 Disturbances

We consider 7 different cases of disturbance. In each case, the disturbance is introduced at the start of the simulation, and as a result, the sequence of disturbances is consistent across all algorithms. The first three cases involve randomly generated disturbances, while the remaining cases involve continuous disturbances, which allows us to study how the algorithms perform when the disturbances are not stochastic. Also, a worst-case disturbance case is considered to assess the performance of the tracking algorithm against an adversary.

- **Uniformly sampled disturbance** It is considered the disturbance to be uniformly sampled from the interval $[0, 1]$.

- **Constant disturbance** The constant disturbance is considered as $w_{1k} = w_{2k} = 1$.

- **Amplitude modulation disturbance** The disturbance is considered as $w_{1k} = w_{2k} = \sin(6\pi k/500)\sin(8\pi k/500)$.

- **Sinusoidal disturbance** A sinusoidal disturbance is considered as $w_{1k} = w_{2k} = \sin(8\pi k/100)$.

- **Gaussian disturbance** Gaussian disturbances is utilized where $w_{1k} \sim \mathcal{N}(0, 0.01)$ and $w_{2k} \sim \mathcal{N}(0, 0.01)$ in this study. If the system's dynamic is known, the optimal controller for an LQR cost is a linear one (Bertsekas, 2012). The support for the Gaussian noise is not finite for the LQR method. That said, the theoretical results require the disturbance to be bounded. However, the actual bound does not necessarily need to be known and could be large, in contrast to robust control methods such as $H_\infty$. In this paper's simulation, the Gaussian noise generator (numpy.random.normal) is utilized which is provided by Numpy in Python that generates bounded samples.

- **Random walk disturbance** It is assumed that the disturbance follows a random walk and is generated by $w_k = 0.999w_{k-1} + \eta_{k-1}$, where $\eta_{k-1} \sim \mathcal{N}(\mathbf{0}, 0.01)$. The internal dynamics of the random walk is chosen to be 0.999 instead of 1 in order to ensure the boundedness of the disturbance. When the noise follows a random walk, the optimal LQR controller is linear. To illustrate this, the random walk disturbance in equation 1 is replaced with

  $$x_{k+1} = Ax_k + Bu_k + 0.999w_{k-1} + \eta_{k-1}.$$

  Here, in each time step $k$, the state $x_k$ is measured, and according to Assumption 1, $w_{k-1}$ is known. A new state variable $\bar{x}k = [x_k^T, w_{k-1}^T]$ is introduced, then one can obtain

  $$\bar{x}_{k+1} = \begin{bmatrix} A & 0.999I \\ \mathbf{0} & 0.999I \end{bmatrix} \bar{x}_k + \begin{bmatrix} B \\ \mathbf{0} \end{bmatrix} u + \begin{bmatrix} I & \mathbf{0} \\ \mathbf{0} & I \end{bmatrix} \eta_{k-1}. \tag{21}$$

  Thus, equation 1 with a random walk disturbance can be viewed as an extended system described by equation 21, where the noise $\eta_{k-1}$ is Gaussian. Consequently, the optimal controller for this extended system is the LQR.

- **Worst-case disturbance (Adversary)** Tracking an unknown reference signal with an adversary agent can be formulated as a two-player zero-sum game in which the control policy seeks to minimize the value function, while the disturbance policy $w_k$ desires to maximize it. The goal is to find the

feedback saddle point $(u_k^*, w_k^*)$ such that if we take the rolling cost $c_i = x_i^T Q x_i + u_i^T R u_i - \gamma^2 w_i^T w_i$, one has

$$J^*(x_k) = \min_{u_k} \max_{w_k} \sum_{i=k}^{T} [e_i^T Q e_i + u_i^T R u_i - \gamma^2 w_i^T w_i]. \tag{22}$$

The worst disturbance based on the formulation of the zero-sum game in (Kiumarsi et al., 2017) can be computed at each state as $w_k^* = -K_w(x_k - r_k)$ where $K_w$ can be computed as

$$K_w = (I^T P I - \gamma^2 I - D^T P B (R + B^T P B)^{-1})^{-1} (I^T P A - I^T P B (R + B^T P B)^{-1} B^T P A)$$

where $P$ satisfies the game algebraic Riccati equation (GARE)

$$P = A^T P A + Q - [A^T P B \ A^T P I] \begin{bmatrix} R + B^T P^B & B^T P D \\ D^T P B & D^T P D - \gamma^2 I \end{bmatrix}^{-1} \begin{bmatrix} B^T P A \\ D^T P A \end{bmatrix}.$$

### 5.3 The compared control approaches

The effectiveness of the presented online tracking algorithm is shown by comparing it to other linear control methods, including the $LQR$ and $H_\infty$ approaches. These approaches optimize a quadratic performance index and are considered optimal for Gaussian and worst-case disturbances, making them the best performers in scenarios where these types of disturbances are present.

In the LQR approach, we consider two cases where the actual model of the system is known, as well as when an estimated model is used for the design. We set $T_0 = 464$, $\lambda = 5$ when we identify the dynamics of the system in an approach. The details are given in the description of each algorithm. In the case of Adversarial disturbance, the identified model from the Gaussian noise is used since the adversarial disturbance makes the open-loop system unstable. Learning a system model despite an adversarial disturbance is a daunting challenge and an open problem.

- **Online state tracking in Algorithm 1:** During the execution of the algorithm, the value of $K$ is maintained unchanged, which can be obtained based on a priori knowledge of the systems' dynamics. Note that this $K$ can be any stabilizing controller that the algorithm is assumed to have access to. The other parameters are chosen as $H = 5$, $m_r = 5$, $m_w = 5$, and $\eta = 0.0001$, and $M$ and $P$ are initialized as zero matrices. In this algorithm, no information about the dynamics of the reference signal is needed. This algorithm only relies on the measured outputs of the reference signal $r_k$. Similarly, no information about the disturbance is Incorporated into this algorithm.

- **LQR:** The feedback controller gain $K_{fb}$ is chosen as $-(R + B^T P_r B)^{-1} B^T P_r A$, where $P_r$ is calculated using $ARE(A, B, Q, R)$, assuming knowledge of the dynamics of the reference signal. Subsequently, $K_{ff}$ is calculated using the approach mentioned in Theorem 1. The control law $u_k = K_{fb} x_k + K_{ff} z_k$ requires knowledge of the state of the reference signal, denoted as $z_k$, which is constructed from $r_k$ using the dynamics of the reference signal as described in Lemma 1 of (Yaghmaie & Modares, 2023).

- **Certainty equivalent (C.E.) LQR and LQR for random walk:** The feedback controller gain $K_{fb}$ is chosen as $-(R + \hat{B}^T P_r \hat{B})^{-1} \hat{B}^T P_r \hat{A}$, where $P_r$ is calculated using $ARE(\hat{A}, \hat{B}, Q, R)$, with $(\hat{A}, \hat{B})$ being the identified system. Subsequently, $K_{ff}$ is computed with the method mentioned in Theorem 1. Also, the control law $u_k = K_{fb} x_k + K_{ff} z_k$ requires knowledge of the state of the reference signal, denoted as $z_k$, which is constructed from $r_k$ using the dynamics of the reference signal as described in Lemma 1 of (Yaghmaie & Modares, 2023).

  It is observed in Subsection 5.2 that when the disturbance is a random walk, the system dynamics can be extended according to equation 21. The extended dynamics involve a Gaussian disturbance, and consequently, LQR for the extended dynamics is used as the optimal controller. In this case, the algorithm is referred to as "*LQR for random walk*".

Table 1: The final maximum and normal difference between the identified dynamics of the system $(\hat{A}, \hat{B})$ and the actual one $(A, B)$. "Maximum difference" refers to the maximum difference in the identification of entries of matrices A and B

| Disturbance | Max Difference for A | Norm of Difference for A | Max Difference for B | Norm of Difference for B |
|---|---|---|---|---|
| Constant | 0.14 | 0.27 | 0.03 | 0.04 |
| Amplitude mod. | 0.11 | 0.17 | 0.06 | 0.08 |
| Sinusoidal | 0.16 | 0.27 | 0.03 | 0.04 |
| Gaussian | 0.07 | 0.07 | 0.03 | 0.03 |
| Random walk | 0.22 | 0.28 | 0.07 | 0.10 |
| Uniformly sam. | 0.06 | 0.08 | 0.02 | 0.03 |

- **Certainty equivalent (C.E.) $H_\infty$-control:** The Certainty equivalent (C.E.) $H_\infty$-control approach aims to design a controller, denoted as $K_{fb}$, for the system described by equation 1, such that the $\mathcal{L}_2$-norm of the system's output, scaled by $\sqrt{Q}$, divided by the $\mathcal{L}2$-norm of the worst-case disturbance input, denoted as $w$, is less than or equal to a threshold. For the sake of comparison, it is assumed that the dynamics of the reference input, described by equation 20, are known. This knowledge is utilized to construct $z_k$ from $r_k$ as described in Lemma 1 of (Yaghmaie & Modares, 2023). Then, the control input $u_k$ is calculated as $u_k = K_{fb}x_k + K_{ff}z_k$, where $K_{ff}$ is the feedforward controller computed with the method mentioned in Theorem 1. Certainty equivalent $H_\infty$-control approach is conservative, as it ensures a finite $\mathcal{L}_2$-gain for the worst-case disturbance.

### 5.4 Evaluation of the identification algorithm

In this subsection, the performance of the identification algorithm is discussed for the 6 cases of the disturbance in Subsection 5.2. In Table 1, the maximum difference between the actual system and the identified one is summarized, as well as the norm of their difference. When the noise is non-Gaussian, system identification using random binary inputs and the least squares method have different implications. Random binary inputs can provide diverse frequency content for analysis, but they may be more sensitive to outliers and non-linear distortions, resulting in potentially less accurate parameter estimates. Meanwhile, the least squares method assumes Gaussian noise, and when this assumption is violated, the parameter estimates obtained may be biased or less accurate due to the influence of non-Gaussian noise. Thus, the choice between these methods should consider the specific non-Gaussian characteristics of the noise to ensure reliable system identification results. In table 1 the results gained from the presented system identification show that even though the disturbances are not Gaussian except for one case, the identification errors are very close to it, with a maximum $\mathcal{L}_2$ norm error of 0.28 for random walk noise and a minimum $\mathcal{L}_2$ norm error of 0.03 for Gaussian noise.

### 5.5 Evaluation of the Tracking Algorithm

In Fig. 1, the reference signal is plotted over $t = [9900, 10000]$ for the representation purpose. The algorithms in Subsection 5.3 are run for $T = 10000$ steps and the final average costs are brought in Table 2. The performance of the algorithms over $t = [9900, 10000]$ is depicted in Fig. 2 - 8.

When the disturbance follows a non-Gaussian or non-random walk distribution, there is no analytical approach to determine the optimal linear feedback policy. In such cases, the $H_\infty$-controller is commonly employed to design a linear feedback policy that ensures a finite $\mathcal{L}_2$-gain for the worst-case disturbance, albeit with a conservative approach. If the actual disturbance is not the worst-case scenario, the $H_\infty$-controller may not yield the best performance. According to Fig. 2 - 7, the presented algorithm has an even better performance in constant, amplitude modulation, sinusoidal, random walk, uniformly distributed, and adversarial disturbances. Additionally, the performance of the presented algorithm is comparable to the actual LQR and certainty-equivalent LQR when Gaussian noise is present.

In the case where the disturbance is Gaussian and the dynamics of the reference signal and the actual system are known, the optimal linear feedback policy can be determined by selecting $K_{fb} = -(R + B^T P_r B)^{-1} B^T P_r A$, where $P_r = ARE(A, B, Q, R)$, and subsequently calculating $K_{ff}$. For the sake of the

Table 2: The final average cost, as introduced in equation 6, incurred by the different algorithms over a duration of T = 10000 steps is presented. Notably, the most competitive average cost values, indicated in bold, are reported for each respective disturbance case. It is noteworthy that the evaluation of LQR for random walk is solely applicable to scenarios involving random walk disturbances, and thus its performance is only assessed in such instances. C.E. refers to certainty equivalent and R.W. refers to random walk.

| Disturbance | Algorithm 1 | C.E. LQR | C.E. $H_\infty$ | C.E./Actual LQR R.W. | LQR |
|---|---|---|---|---|---|
| Constant | **8.04** | 40.07 | 29.18 | N.A. | 57.76 |
| Amplitude mod. | **7.83** | 16.74 | 12.72 | N.A. | 17.58 |
| Sinusoidal | **15.21** | 27.71 | 20.08 | N.A. | 30.21 |
| Gaussian | 5.62 | 5.32 | 5.29 | N.A. | **5.25** |
| Random walk | 17.75 | 236.11 | 163.02 | 19.22 / **15.48** | 236.68 |
| Uniformly sam. | **10.09** | 19.51 | 16.09 | N.A. | 21.99 |
| Adversarial | **13.76** | 17.21 | 14.75 | N.A. | 17.17 |

experiment, one can take $K_{fb}$ computed this way as the initial stabilizing control gain for the presented algorithm as well. The results can be seen in Table 2. The average cost of the presented tracking approach is considerably close to the optimal control when the noise is Gaussian (LQR). This can be seen in Figure 9. A similar discourse is applicable to the scenario of a random-walk disturbance, as elucidated in Subsection 5.2, where it is demonstrated that the optimal controller for the system in the presence of a random-walk disturbance can be obtained by solving an LQR problem for the extended system. That said, the uncertainty that is present in the model identification of the system resulted in poorer performance of the certainty equivalents as compared to the presented algorithm.

# 6 Conclusion

In this paper, the challenge of state tracking in the presence of general disturbances is addressed, even when the dynamics of the actual system are unknown. An algorithm that combines an identification period with a memory-augmented robust tracking algorithm is introduced. This presented algorithm enables online tuning of the controller parameters to achieve state tracking and disturbance rejection while minimizing convex costs. It is shown that the presented online algorithm achieves a policy regret of $\mathcal{O}(T^{2/3})$. In our future research, we plan to extend our approach to partially observable dynamical systems and eliminate the bounded assumption on the reference signal.

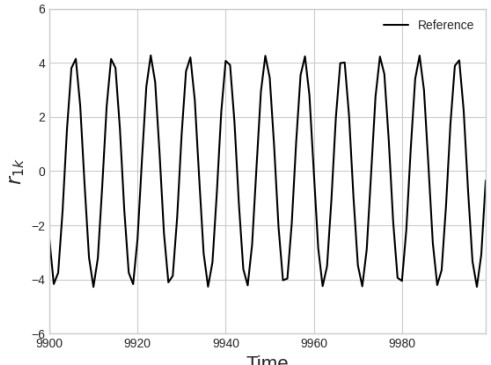 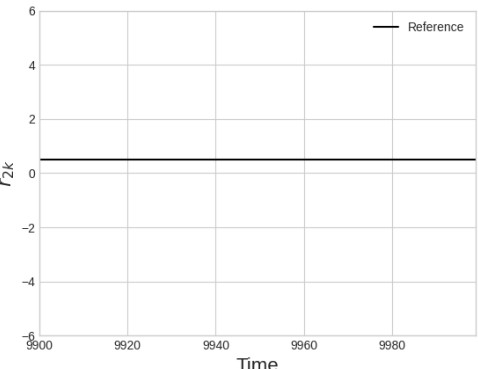

Figure 1: The reference signals used for the evaluation and comparison of various control algorithms under different disturbances.

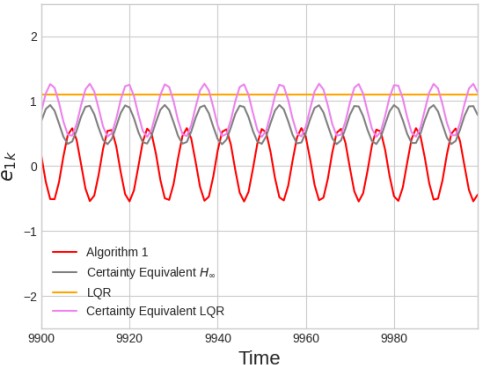 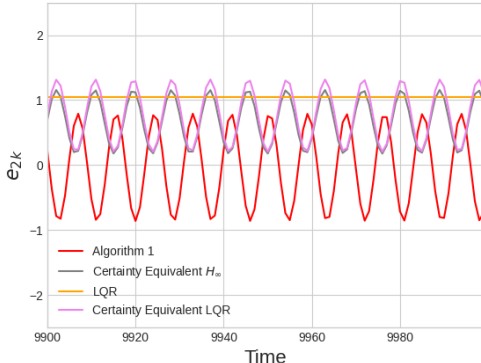

Figure 2: Tracking error for constant disturbance for the presented Algorithm 1, versus certainty equivalent $H_\infty$-control, certainty equivalent LQR control, and LQR control knowing the dynamics of the system using the reference signals in Fig. 1.

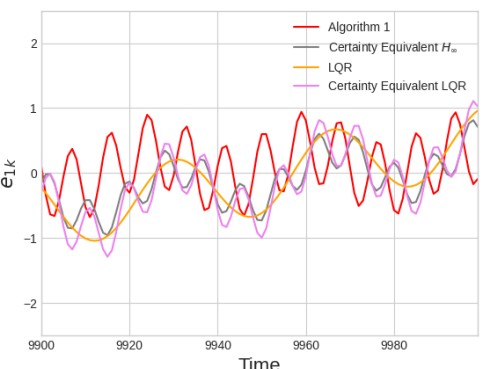 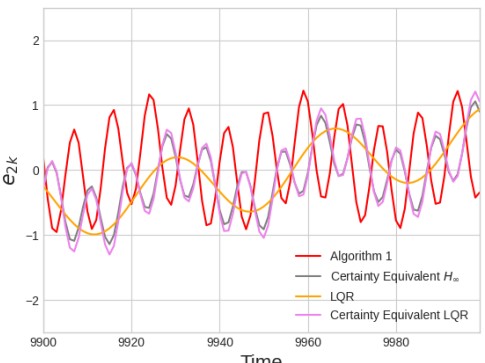

Figure 3: Tracking error for amplitude modulation disturbance for the presented Algorithm 1, versus certainty equivalent $H_\infty$-control, certainty equivalent LQR control, and LQR control knowing the dynamics of the system using the reference signals in Fig. 1.

## Acknowledgment

Nariman Niknejad and Hamidreza Modares are supported by the Department of Navy award N00014-22-1-2159 issued by the Office of Naval Research, USA. Farnaz Adib Yaghmaie is supported by the Excellence Center at Linköping–Lund in Information Technology (ELLIIT), ZENITH, and partially by Sensor informatics and Decision-making for the Digital Transformation (SEDDIT).

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

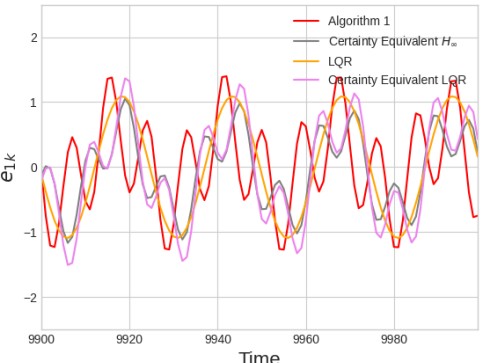 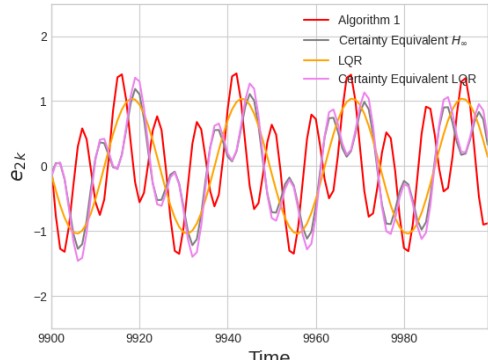

Figure 4: Tracking error for sinusoidal disturbance for the presented Algorithm 1, versus certainty equivalent $H_\infty$-control, certainty equivalent LQR control, and LQR control knowing the dynamics of the system using the reference signals in Fig. 1.

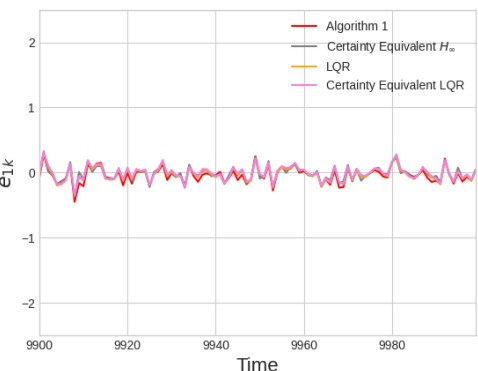 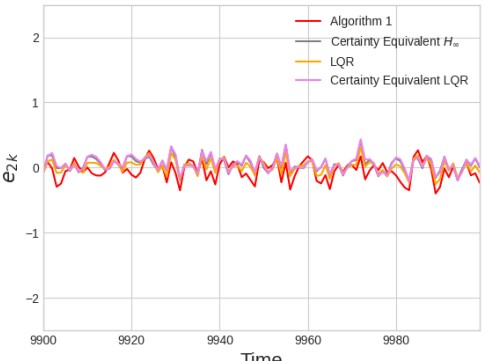

Figure 5: Tracking error for Gaussian disturbance for the presented Algorithm 1, versus the certainty equivalent $H_\infty$-control, certainty equivalent LQR control, and LQR control knowing the dynamics of the system using the reference signals in Fig. 1.

154–165, 2019.

Dimitri Bertsekas. *Dynamic programming and optimal control: Volume I*. Athena scientific, 2012.

Dimitri Bertsekas. *Reinforcement learning and optimal control*. Athena Scientific, 2019.

Ci Chen, Hamidreza Modares, Kan Xie, Frank L. Lewis, Yan Wan, and Shengli Xie. Reinforcement Learning-Based Adaptive Optimal Exponential Tracking Control of Linear Systems with Unknown Dynamics. *IEEE Transactions on Automatic Control*, 64(11):4423–4438, 2019. ISSN 15582523. doi: 10.1109/TAC.2019. 2905215.

Ci Chen, Lihua Xie, Yi Jiang, Kan Xie, and Shengli Xie. Robust output regulation and reinforcement learning-based output tracking design for unknown linear discrete-time systems. *IEEE Transactions on Automatic Control*, pp. 1–1, 2022.

Richard Cheng, Gábor Orosz, Richard M Murray, and Joel W Burdick. End-to-end safe reinforcement learning through barrier functions for safety-critical continuous control tasks. In *Proceedings of the AAAI conference on artificial intelligence*, volume 33, pp. 3387–3395, 2019.

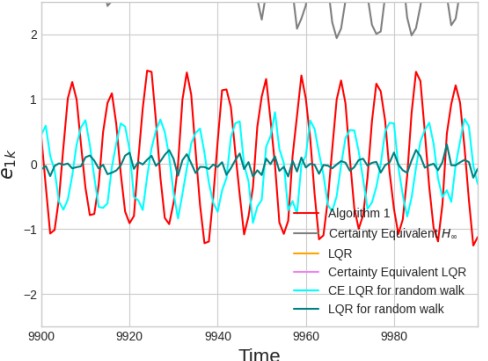 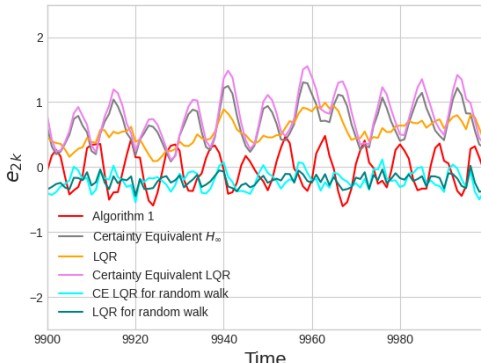

Figure 6: Tracking error for the random walk disturbance for the presented Algorithm 1, versus certainty equivalent $H_\infty$-control, certainty equivalent LQR control, certainty equivalent LQR for random walk, LQR control, and LQR control for random walk disturbance knowing the dynamics of the system using the reference signals in Fig. 1.

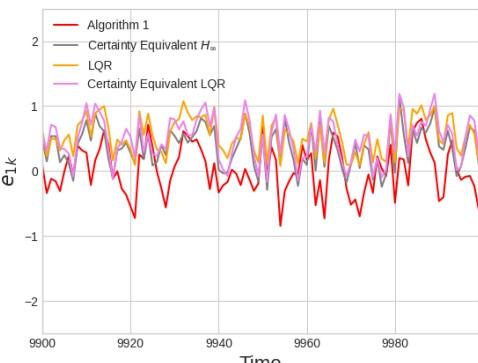 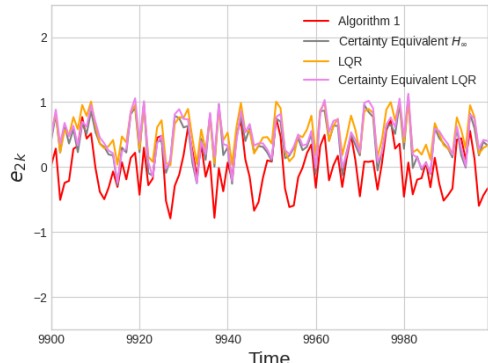

Figure 7: Tracking error for the uniformly sampled disturbance for the presented Algorithm 1, versus certainty equivalent $H_\infty$-control, certainty equivalent LQR control, and the LQR control knowing the dynamics of the system using the reference signals in Fig. 1.

A. Chiuso and G. Pillonetto. System identification: A machine learning perspective. *Annual Review of Control, Robotics, and Autonomous Systems*, 2(1):281–304, 2019. doi: 10.1146/annurev-control-053018-023744. URL https://doi.org/10.1146/annurev-control-053018-023744.

Alon Cohen, Avinatan Hasidim, Tomer Koren, Nevena Lazic, Yishay Mansour, and Kunal Talwar. Online linear quadratic control. In *International Conference on Machine Learning*, pp. 1029–1038. PMLR, 2018.

Diyako Dadkhah and SO Reza Moheimani. Combining $H_\infty$ and resonant control to enable high-bandwidth measurements with a MEMS force sensor. *Mechatronics*, 96:103086, 2023.

Chao Deng, Xiao-Zheng Jin, Wei-Wei Che, and Hai Wang. Learning-based distributed resilient fault-tolerant control method for heterogeneous mass under unknown leader dynamic. *IEEE Transactions on Neural Networks and Learning Systems*, 33(10):5504–5513, 2021.

Warren E Dixon, Marcio S de Queiroz, Darren M Dawson, and Terrance J Flynn. Adaptive tracking and regulation of a wheeled mobile robot with controller/update law modularity. *IEEE Transactions on control systems technology*, 12(1):138–147, 2004.

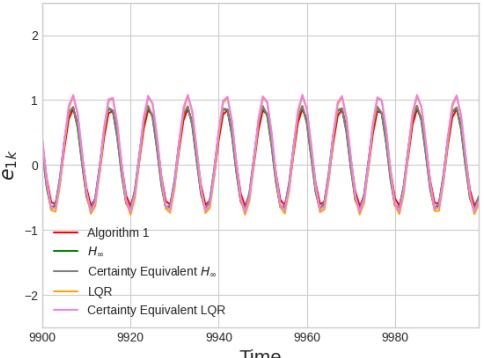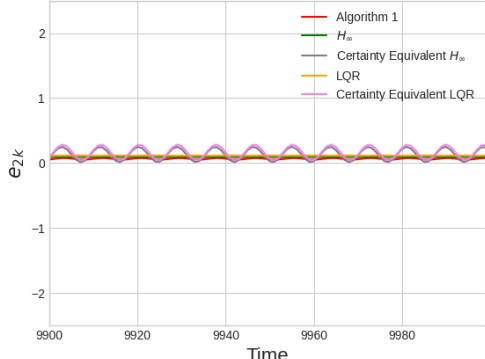

Figure 8: Tracking error for the adversary disturbance for the presented Algorithm 1, versus the $H_\infty$-control, certainty equivalent $H_\infty$-control, certainty equivalent LQR control, and LQR control knowing the dynamics of the system using the reference signals in Fig. 1.

John Doyle. Robust and optimal control. *Proceedings of 35th IEEE Conference on Decision and Control*, 2: 1595–1598 vol.2, 1995.

Quan-Yong Fan, Yucong Sun, and Bin Xu. Improved data-driven control design based on LMI and its applications in lithium-ion batteries. *IEEE Transactions on Circuits and Systems II: Express Briefs*, 2023.

Mohamad Kazem Shirani Faradonbeh, Ambuj Tewari, and George Michailidis. Finite time identification in unstable linear systems, 2017. URL https://arxiv.org/abs/1710.01852.

Aleksandra Faust, Nick Malone, and Lydia Tapia. Preference-balancing motion planning under stochastic disturbances. In *2015 IEEE International Conference on Robotics and Automation (ICRA)*, pp. 3555–3562. IEEE, 2015.

Weinan Gao and Zhong Ping Jiang. Global Optimal Output Regulation of Partially Linear Systems via Robust Adaptive Dynamic Programming. *IFAC-PapersOnLine*, 48(11 11):742–747, 2015. ISSN 24058963. doi: 10.1016/j.ifacol.2015.09.278. URL http://dx.doi.org/10.1016/j.ifacol.2015.09.278.

Weinan Gao and Zhong Ping Jiang. Adaptive Dynamic Programming and Adaptive Optimal Output Regulation of Linear Systems. *IEEE Transactions on Automatic Control*, 61(12):4164–4169, 2016. ISSN 00189286. doi: 10.1109/TAC.2016.2548662.

Weinan Gao and Zhong-Ping Jiang. Learning-based adaptive optimal output regulation of linear and nonlinear systems: an overview. *Control Theory and Technology*, 20(1):1–19, 2022.

Weinan Gao, Zhong-Ping Jiang, Frank L. Lewis, and Yebin Wang. Cooperative optimal output regulation of multi-agent systems using adaptive dynamic programming. In *2017 American Control Conference (ACC)*, pp. 2674–2679, 2017.

Gaofeng Hao, Zhuang Fu, Xin Feng, Zening Gong, Peng Chen, Dan Wang, Weibin Wang, and Yang Si. A deep deterministic policy gradient approach for vehicle speed tracking control with a robotic driver. *IEEE Transactions on Automation Science and Engineering*, 19(3):2514–2525, 2021.

Elad Hazan, Sham M. Kakade, and Karan Singh. The Nonstochastic Control Problem. In *Algorithmic Learning Theory*, pp. 408—-421, 2020. URL http://arxiv.org/abs/1911.12178.

Jie Huang. *Nonlinear output regulation: theory and applications*. SIAM, 2004.

Alberto Isidori. *Nonlinear control systems: an introduction*. Springer, 1985.

Yi Jiang, Bahare Kiumarsi, Jialu Fan, Tianyou Chai, Jinna Li, and Frank L. Lewis. Optimal Output Regulation of Linear Discrete-Time Systems with Unknown Dynamics Using Reinforcement Learning. *IEEE Transactions on Cybernetics*, 50(7):3147–3156, 2020a. ISSN 21682275. doi: 10.1109/TCYB.2018. 2890046.

Yi Jiang, Bahare Kiumarsi, Jialu Fan, Tianyou Chai, Jinna Li, and Frank L. Lewis. Optimal output regulation of linear discrete-time systems with unknown dynamics using reinforcement learning. *IEEE Transactions on Cybernetics*, 50(7):3147–3156, 2020b.

Hassan K. Khalil. *Nonlinear Systems*. Prentice Hall, second edition, 2002.

Bahare Kiumarsi, Frank L. Lewis, Hamidreza Modares, Ali Karimpour, and Mohammad Bagher Naghibi-Sistani. Reinforcement Q-learning for optimal tracking control of linear discrete-time systems with unknown dynamics. *Automatica*, 50(4):1167–1175, 2014. ISSN 00051098. doi: 10.1016/j.automatica.2014. 02.015. URL http://dx.doi.org/10.1016/j.automatica.2014.02.015.

Bahare Kiumarsi, Frank L Lewis, and Zhong-Ping Jiang. $H_\infty$ control of linear discrete-time systems: Off-policy reinforcement learning. *Automatica*, 78:144–152, 2017.

Frank. L. Lewis. Optimal control((book)). *New York, Wiley-Interscience, 1986, 371*, 1986.

Bohao Li and Yunjie Wu. Path planning for uav ground target tracking via deep reinforcement learning. *IEEE access*, 8:29064–29074, 2020.

Hongyi Li, Ying Wu, and Mou Chen. Adaptive fault-tolerant tracking control for discrete-time multiagent systems via reinforcement learning algorithm. *IEEE Transactions on Cybernetics*, 51(3):1163–1174, 2020.

Lennart Ljung. *System identification toolbox: User's guide*. Citeseer, 1995.

Lennart Ljung. *System identification*. Springer, 1998.

Alireza Mehrzad, Milad Darmiani, Yashar Mousavi, Miadreza Shafie-Khah, and Mohammadreza Aghamohammadi. A review on data-driven security assessment of power systems: Trends and applications of artificial intelligence. *IEEE Access*, 2023.

Di Mei, Jian Sun, Lihua Dou, and Yong Xu. Learning-based distributed adaptive control of heterogeneous multi-agent systems with unknown leader dynamics. *IET Cyber-Physical Systems: Theory & Applications*, 2022.

Hamidreza Modares, Frank L. Lewis, and Zhong Ping Jiang. $H_\infty$ Tracking Control of Completely Unknown Continuous-Time Systems via Off-Policy Reinforcement Learning. *IEEE Transactions on Neural Networks and Learning Systems*, 26(10):2550–2562, 2015. ISSN 21622388. doi: 10.1109/TNNLS.2015.2441749.

Hamidreza Modares, Subramanya P Nageshrao, Gabriel A Delgado Lopes, Robert Babuška, and Frank L Lewis. Optimal model-free output synchronization of heterogeneous systems using off-policy reinforcement learning. *Automatica*, 71:334–341, 2016.

Mehdi Mohammadi, Mohammad Mehdi Arefi, Peyman Setoodeh, and Okyay Kaynak. Optimal tracking control based on reinforcement learning value iteration algorithm for time-delayed nonlinear systems with external disturbances and input constraints. *Information Sciences*, 554:84–98, 2021.

J. Nagumo and A. Noda. A learning method for system identification. *IEEE Transactions on Automatic Control*, 12(3):282–287, 1967. doi: 10.1109/TAC.1967.1098599.

Antonin Novak, Laurent Simon, František Kadlec, and Pierrick Lotton. Nonlinear system identification using exponential swept-sine signal. *IEEE Transactions on Instrumentation and Measurement*, 59(8):2220–2229, 2009.

Pedram Rabiee and Amirsaeid Safari. Safe exploration in reinforcement learning: Training backup control barrier functions with zero training time safety violations, 2023.

Tuhin Sarkar, Alexander Rakhlin, and Munther A. Dahleh. Nonparametric finite time lti system identification, 2019. URL https://arxiv.org/abs/1902.01848.

Richard S Sutton and Andrew G Barto. *Reinforcement learning: An introduction.* MIT press, 2018.

Farzaneh Tatari, Majid Mazouchi, and Hamidreza Modares. Fixed-time system identification using concurrent learning. *IEEE Transactions on Neural Networks and Learning Systems*, pp. 1–11, 2021. doi: 10.1109/TNNLS.2021.3125145.

Kyriakos G Vamvoudakis, Arman Mojoodi, and Henrique Ferraz. Event-triggered optimal tracking control of nonlinear systems. *International Journal of Robust and Nonlinear Control*, 27(4):598–619, 2017.

Licheng Wang, Engang Tian, Changsong Wang, and Shuai Liu. Secure estimation against malicious attacks for lithium-ion batteries under cloud environments. *IEEE Transactions on Circuits and Systems I: Regular Papers*, 69(10):4237–4247, 2022a.

Ning Wang, Ying Gao, Chen Yang, and Xuefeng Zhang. Reinforcement learning-based finite-time tracking control of an unknown unmanned surface vehicle with input constraints. *Neurocomputing*, 484:26–37, 2022b.

Thomas Weber, Johannes Sossenheimer, Steffen Schäfer, Moritz Ott, Jessica Walther, and Eberhard Abele. Machine learning based system identification tool for data-based energy and resource modeling and simulation. *Procedia CIRP*, 80:683–688, 2019. ISSN 2212-8271. doi: https://doi.org/10.1016/j.procir.2018.12.021. URL https://www.sciencedirect.com/science/article/pii/S2212827118313003. 26th CIRP Conference on Life Cycle Engineering (LCE) Purdue University, West Lafayette, IN, USA May 7-9, 2019.

Farnaz Adib Yaghmaie and Hamidreza Modares. Online optimal tracking of linear systems with adversarial disturbances. *Transactions on Machine Learning Research*, 2023. ISSN 2835-8856. URL https://openreview.net/forum?id=5nVJlKgmxp.

Zhikai Yao and Jianyong Yao. Toward reliable designs of data-driven reinforcement learning tracking control for euler–lagrange systems. *Neural Networks*, 153:564–575, 2022.

Soroush Zare, Mohammad Reza Hairi Yazdi, Mehdi Tale Masouleh, Dan Zhang, Sahand Ajami, and Amirhossein Afkhami Ardekani. Experimental study on the control of a suspended cable-driven parallel robot for object tracking purpose. *Robotica*, 40(11):3863–3877, 2022.

Hongwei Zhang and Frank L Lewis. Adaptive cooperative tracking control of higher-order nonlinear systems with unknown dynamics. *Automatica*, 48(7):1432–1439, 2012.

# A   Real-world Application Example

In the main body of the paper, we compared the performance of the presented algorithm with LQR, $H_\infty$, and their certainty equivalent versions. In this subsection, the implementation of the algorithm is brought on a real-world application. The operational efficiency of lithium-ion batteries is inherently influenced by various factors in real-world situations, such as ambient temperature, battery aging, and operational status. These factors pose significant risks to the secure functioning of lithium batteries and may potentially lead to undesirable damage to the connected equipment. Consider the following tracking a constant signal problem where the dynamics of the system of a Lithium-ion battery from (Fan et al., 2023; Wang et al., 2022a) as shown in Figure 10 is

$$x_{k+1} = \begin{bmatrix} 1 - \frac{1}{R_1 C_1} & 0 & 0 \\ 0 & 1 - \frac{1}{R_2 C_2} & 0 \\ 0 & 0 & 1 \end{bmatrix} x_k + \begin{bmatrix} \frac{1}{C_1} \\ \frac{1}{C_2} \\ -\frac{1}{C_b} \end{bmatrix} u_k + w_k, \tag{23}$$

where nominal values are $(C_1, C_2, C_b) = (39.91\,F, 3.50\,F, 40\,F)$ and $(R_1, R_2) = (0.0269\,\Omega, 7.3\,\Omega)$ and the reference signal is generated by

$$z_{k+1} = \begin{bmatrix} 1 & 0 & 0 \\ 0 & 1 & 0 \\ 0 & 0 & 1 \end{bmatrix} z_k, \ z_0 = [-0.02, -5.20, 0.89]^T,$$

$$r_k = \begin{bmatrix} 1 & 0 & 0 \\ 0 & 1 & 0 \\ 0 & 0 & 1 \end{bmatrix} z_k. \tag{24}$$

where

$$x_k = \begin{bmatrix} x_{1k} \\ x_{2k} \\ x_{3k} \end{bmatrix}, \ w_k = \begin{bmatrix} w_{1k} \\ w_{2k} \\ w_{3k} \end{bmatrix}, \ r_k = \begin{bmatrix} r_{1k} \\ r_{2k} \\ r_{3k} \end{bmatrix}, \ e_k = \begin{bmatrix} e_{1k} \\ e_{2k} \\ e_{3k} \end{bmatrix} = \begin{bmatrix} x_{1k} - r_{1k} \\ x_{2k} - r_{2k} \\ x_{3k} - r_{3k} \end{bmatrix}.$$

A quadratic cost with $Q = I_3$, $R = 1$ is considered; that is

$$c_k = e_k^T Q e_k + u_k^T R u_k.$$

## A.1   Disturbances

Same as the previous simulation, in this section we brought the comparison of the different control algorithms implemented on the system under different disturbances.

- **Uniformly sampled disturbance** It is considered the disturbance to be uniformly sampled from the interval $[0, 0.5]$.

- **Constant disturbance** The constant disturbance is considered as $w_{1k} = w_{2k} = w_{3k} = 0.05$.

- **Amplitude modulation disturbance** The disturbance is considered as $w_{1k} = w_{2k} = w_{3k} = 0.05 \times \sin(6\pi k/500)\sin(8\pi k/500)$.

- **Sinusoidal disturbance** A sinusoidal disturbance is considered as $w_{1k} = w_{2k} = w_{3k} = 0.05 \times \sin(8\pi k/100)$.

- **Gaussian disturbance** Gaussian disturbances is utilized where $w_{1k} \sim \mathcal{N}(0, 0.05)$ and $w_{2k} \sim \mathcal{N}(0, 0.05)$ in this case.

- **Worst-case disturbance (Adversary)** Same as explained in Section 5, the worst-case disturbance is generated using the zero-sum game formulation.

Table 3: The final maximum and normal difference between the identified dynamics of the system $(\hat{A}, \hat{B})$ and the actual one $(A, B)$. "Maximum difference" refers to the maximum difference in the identification of entries of matrices A and B

| Disturbance | Max Difference for A | Norm of Difference for A | Max Difference for B | Norm of Difference for B |
|---|---|---|---|---|
| Constant | 0.89 | 0.89 | 0.00 | 0.00 |
| Amplitude mod. | 0.09 | 0.11 | 0.00 | 0.00 |
| Sinusoidal | 0.19 | 0.25 | 0.00 | 0.00 |
| Gaussian | 0.32 | 0.45 | 0.00 | 0.00 |
| Uniformly sam. | 0.65 | 0.68 | 0.27 | 0.27 |

## A.2 The compared control approaches

same as the previous simulation example, the below control algorithms are employed to assess their performance against the previously mentioned disturbances.

- Online state tracking in Algorithm 1:During the execution of the algorithm, the value of $K$ is maintained unchanged, which can be obtained based on a priori knowledge of the systems' dynamics. Note that this $K$ can be any stabilizing controller that the algorithm is assumed to have access to. The other parameters are chosen as $H = 5$, $m_r = 5$, $m_w = 5$, and $\eta = 0.0001$, and $M$ and $P$ are initialized as zero matrices.

- LQR with the actual system matrices

- Certainty equivalent (C.E.) LQR

- Certainty equivalent (C.E.) $H_\infty$-control

The settings of the last three algorithms are the same as defined before.

## A.3 Evaluation of the identification algorithm

In this subsection, the performance of the identification algorithm is discussed for the 5 cases of the disturbance mentioned in the previous section. The identification period for this system is 293. In Table 3, the maximum difference between the actual system and the identified one is summarized, as well as the $\mathcal{L}_2$ norm of their difference. As can be seen in Table 3, the performance of the system identification is the worst for the "Constant disturbance" case with $\mathcal{L}_2(A - \hat{A}) = 0.89$ which is similar to the results of the previous example. The best identification accuracy happened for the "Amplitude modulation" case.

## A.4 Evaluation of the Tracking Algorithm

for this example, the algorithms are run for $T = 5000$ steps and the final average costs are brought in Table 4. Tracking the constant signals for the three states of the system $t = [4900, 5000]$ is depicted in Fig. 11 - 16. As it can be seen, due to the inherent limitations in computing the feedforward term during tracking, particularly when matrix B lacks square dimensions, the accuracy of the feedforward term is compromised. Consequently, across all algorithms reliant on this computation such as C.E. $LQR$ and $H_\infty$, a discernible decline in tracking performance becomes evident, emphasizing the superiority of the presented algorithm. It can be seen in Table 4, that the presented algorithm outperforms all other methods in the tracking of the defined reference signal.

Table 4: The final average cost, as introduced in equation 6, incurred by the different algorithms over a duration of $T = 5000$ steps is presented. Notably, the most competitive average cost values, indicated in bold, are reported for each respective disturbance case.

| Disturbance | Algorithm 1 | C.E. LQR | C.E. $H_\infty$ | LQR |
|---|---|---|---|---|
| Constant | **2.35** | 28.28 | 51.15 | 30.77 |
| Amplitude mod. | **1.34** | 23.32 | 2.02 | 22.80 |
| Sinusoidal | **1.17** | 27.36 | 20.94 | 22.63 |
| Gaussian | **1.19** | 30.51 | 70.02 | 22.63 |
| Uniformly sam. | **1.94** | 31.45 | 26.01 | 23.61 |
| Adversarial | **1.77** | 12.66 | 2.60 | 7.58 |

**Appendix B: Theoretical Results**

## B    Results related to the memory augmented controller

In this section, a list of results related to the memory-augmented controllers is brought.

- Lemma 3 provides bounds for $\Psi_{k,y}^{K,h}$ and $\psi_{k,z}^{K,h}$.

- Lemma 4 gives bounds on the states and inputs.

- Lemma 5 provides a bound on the tracking error.

- Lemma 6 defines the Lipschitz condition on the truncated cost.

- Lemma 7 gives a bound on the gradient of the truncated cost for the tracking algorithm.

Lemmas 3, 6-7 are given in (Yaghmaie & Modares, 2023) and Lemmas 4-5 are inspired by (Yaghmaie & Modares, 2023) and are essential in proving the main results in Theorem 4.

**Lemma 3 (Lemma 4 in (Yaghmaie & Modares, 2023))** *Let Assumptions 1-5 hold. Suppose that $K$ is $(\kappa, \gamma)$-strongly stable. Then,*

$$
\begin{aligned}
&\|\Psi_{k,y}^{K,h}\| \leq \kappa^2 (1-\gamma)^y \mathbb{I}_{y \leq h-1} + m_w \kappa^5 \kappa_b^2 (1-\gamma)^{y-1}, \\
&\|\psi_{k,z}^{K,h}\| \leq m_r \kappa^5 \kappa_b^2 (1-\gamma)^{z-1}.
\end{aligned}
\tag{25}
$$

**Lemma 4** *Let Assumptions 1-5 hold. Define*

$$
\begin{aligned}
Y_{0:k} &:= [M_{0:k}, P_{0:k}], \\
Y_{H,k} &:= [M_{k-H:k}, P_{k-H:k}].
\end{aligned}
$$

$$
D := \gamma^{-1} \frac{\kappa_w \kappa^3 + (\kappa_r m_r + \kappa_w m_w)(1-\gamma)^{-1} \kappa^6 \kappa_b^2}{1 - \kappa^2 (1-\gamma)^H} + \frac{(\kappa_w + \kappa_b \kappa \kappa_z)\kappa_b \kappa^3}{\gamma},
$$

*Suppose that $K$ and $K_{fb}^*$ are $(\kappa, \gamma)$-strongly stable. Define $x_k^{lin}(K_{fb}^*, K_{ff}^*)$ as the system state corresponding to an optimal linear feedback controller. Then, one has*

$$
\begin{aligned}
&\max(\|x_k^\pi(Y_{0:k-1})\|, \|\tilde{x}_k^\pi(Y_{H,k-1})\|, \|x_k^{lin}(K_{fb}^*, K_{ff}^*)\|) \leq D, \\
&\max(\|u_k^\pi(Y_{0:k})\|, \|\tilde{u}_k^\pi(Y_{H,k})\|) \leq D, \\
&\|x_k^\pi(Y_{0:k-1}) - \tilde{x}_k^\pi(Y_{H,k-1})\| \leq \kappa^2 (1-\gamma)^H D, \\
&\|u_k^\pi(Y_{0:k}) - \tilde{u}_k^\pi(Y_{H,k})\| \leq \kappa^3 (1-\gamma)^H D.
\end{aligned}
\tag{26}
$$

*Proof:* Using equation 11, one has

$$\|x_k^\pi\| \le \|\tilde{A}_K^H\|\|x_{k-H}^\pi\| + \kappa_w \sum_{y=0}^{m_w+H-1} \|\Psi_{k,y}^{K,H}(M_{k-H-1:k-1})\| + \kappa_r \sum_{z=0}^{m_r+H-1} \|\psi_{k,z}^{K,H}(P_{k-H-1:k-1})\|$$

$$\le \kappa^2(1-\gamma)^H\|x_{k-H}^\pi\| + \kappa_w\gamma^{-1}(\kappa^2 + m_w\kappa^5\kappa_b^2(1-\gamma)^{-1}) + \kappa_r\gamma^{-1}(m_r\kappa^5\kappa_b^2)(1-\gamma)^{-1}.$$

The above recursion satisfies

$$\|x_k^\pi\| \le \gamma^{-1}\frac{\kappa_w\kappa^2 + (\kappa_r m_r + \kappa_w m_w)(1-\gamma)^{-1}\kappa^5\kappa_b^2}{1-\kappa^2(1-\gamma)^H}.$$

Similarly, from equation 13, one has

$$\|\tilde{x}_k^\pi(Y_{H,k-1})\| \le \sum_{y=0}^{m_w+H-1} \|\Psi_{k,y}^{K,H}(M_{k-H-1:k-1})w_{k-y-1}\| + \sum_{z=0}^{m_r+H-1} \|\psi_{k,z}^{K,H}(P_{k-H-1:k-1})r_{k-z}\|$$

$$\le \gamma^{-1}\kappa_w\kappa^2 + \gamma^{-1}(\kappa_w m_w + \kappa_r m_r)\kappa^5\kappa_b^2(1-\gamma)^{-1} \le D.$$

where the last inequality is obtained because $0 \le 1 - \kappa^2(1-\gamma)^H \le 1$. Moreover,

$$\|x_k^{lin}(K_{fb}^*, K_{ff}^*)\| = \|\sum_{y=0}^{k-1} \tilde{A}_{K_{fb}^*}^y w_{k-y-1} + \sum_{i=0}^{k-1} \tilde{A}_{K_{fb}^*}^i BK_{ff}^* z_{k-i}\|$$

$$\le \gamma^{-1}\kappa^2(\kappa_w + \kappa\kappa_b\kappa_z) \le D.$$

Besides, one has

$$\|u_k^\pi(Y_{0:k})\| = \|Kx_k^\pi(Y_{0:k-1}) + \sum_{t=1}^{m_w} M^{[t-1]}w_{k-t} + \sum_{s=0}^{m_r-1} P^{[s]}r_{k-s}\|$$

$$\le \kappa\|x_k^\pi(Y_{0:k-1})\| + \kappa_w \sum_{t=1}^{m_w} \kappa_b\kappa^3(1-\gamma)^{(t-1)} + \kappa_r \sum_{s=0}^{m_r-1} \kappa_b\kappa^3(1-\gamma)^s$$

$$\le \gamma^{-1}\frac{\kappa_w\kappa^3 + (\kappa_r m_r + \kappa_w m_w)(1-\gamma)^{-1}\kappa^6\kappa_b^2}{1-\kappa^2(1-\gamma)^H} + \frac{(\kappa_w + \kappa_r)\kappa_b\kappa^3}{\gamma} \le D.$$

Similarly,

$$\|\tilde{u}_k^\pi(Y_{H,k})\| = \|K\tilde{x}_k^\pi(Y_{H,k-1}) + \sum_{t=1}^{m_w} M^{[t-1]}w_{k-t} + \sum_{s=0}^{m_r-1} P^{[s]}r_{k-s}\|$$

$$\le \kappa\|\tilde{x}_k^\pi(Y_{H,k-1})\| + \kappa_w \sum_{t=1}^{m_w} \kappa_b\kappa^3(1-\gamma)^{(t-1)} + \kappa_r \sum_{s=0}^{m_r-1} \kappa_b\kappa^3(1-\gamma)^s$$

$$\le \gamma^{-1}\kappa_w\kappa^3 + \gamma^{-1}(\kappa_w m_w + \kappa_r m_r)\kappa^6\kappa_b^2(1-\gamma)^{-1} + \frac{(\kappa_w + \kappa_r)\kappa_b\kappa^3}{\gamma} \le D.$$

To bound the difference between the actual and truncated state, from equation 13 and equation 11, one has

$$\|x_k^\pi(Y_{0:k-1}) - \tilde{x}_k^\pi(Y_{H,k-1})\| = \|\tilde{A}_K^H x_{k-H}^\pi(Y_{0:k-H-1})\| \le \kappa^2(1-\gamma)^H D,$$

which gives

$$\|u_k^\pi(Y_{0:k}) - \tilde{u}_k^\pi(Y_{H,k})\| \le \|K\|\|\tilde{A}_K^H x_{k-H}^\pi(Y_{0:k-H-1})\| \le \kappa^3(1-\gamma)^H D.$$

This completes the proof.

**Lemma 5** *Let Assumptions 1-5 hold. Suppose that $K$ is $(\kappa, \gamma)$-strongly stable. Define the tracking error as*

$$e_k := x_k^\pi(Y_{0:k-1}) - Fz_k.$$

*Then,*

$$\|e_k\| \leq \kappa_w \gamma^{-1}(\kappa^2 + m_w \kappa^5 \kappa_b^2(1-\gamma)) + \kappa_r \gamma^{-1}(1-\gamma)^{l-1} m_r \kappa^5 \kappa_b^2 + \kappa_r \sum_{z=0}^{l-1} \kappa_z.$$

*Proof:* Without loss of generality and for simplicity, assume that $\|F\| = \|F^{-1}\| \leq \kappa$. The tracking error reads

$$e_k = \sum_{y=0}^{k-1} \Psi_{k,y}^{K,k}(M_{0:k-1})w_{k-y-1} + \sum_{z=0}^{k-1} \psi_{k,z}^{K,k}(P_{0:k-1})r_{k-z} - Fz_k.$$

Using the bounds in equation 25

$$\|e_k\| \leq \sum_{y=0}^{k-1}(\kappa^2(1-\gamma)^y \mathbb{I}_{y \leq k-1} + m_w \kappa^5 \kappa_b^2(1-\gamma)^{y-1})\kappa_w + \kappa \kappa_z + \sum_{z=0}^{k-1} m_r \kappa^5 \kappa_b^2(1-\gamma)^{z-1}\kappa_r$$

$$\leq \kappa_w \gamma^{-1}(\kappa^2 + m_w \kappa^5 \kappa_b^2(1-\gamma)^{-1}) + \kappa \kappa_z + \kappa_r \gamma^{-1} m_r \kappa^5 \kappa_b^2,$$

which is based on the fact that $\sum_{n=0}^{N}(1-\gamma)^n \leq \frac{1}{\gamma}$ in the last inequality.

**Lemma 6 (Lemma 7 in (Yaghmaie & Modares, 2023))** *Let Assumptions 1-5 hold. Define $Y_{H,k} = [Y_1, ..., Y_t, ..., Y_{2H}] = [M_{k-H:k} \ P_{k-H:k}]$ and $\tilde{Y}_{H,k} = [Y_1, ..., \tilde{Y}_t, ..., Y_{2H}]$ where $\tilde{Y}_{H,k}$ has all its elements the same as $Y_{H,k}$, except one element. Then, the truncated cost function in equation 15 satisfies the following Lipschitz condition*

$$|f_k(Y_1, , ..., Y_t, ..., Y_{2H}) - f_k(Y_1, , ..., \tilde{Y}_t, ..., Y_{2H})| \leq L_f \|Y_t - \tilde{Y}_t\|$$

*where*

$$L_f := 3G_c D \, \kappa_b \kappa^3(\kappa_r + \kappa_w).$$

**Lemma 7 (Lemma 8 in (Yaghmaie & Modares, 2023))** *Let Assumptions 1-5 hold. The following gradient bound is satisfied*

$$\|\nabla_{Y_{H,k}} f_k(Y_{H,k})\|_F \leq 6Hd^2 G_c \, (\kappa_r + \kappa_w)\kappa_b \kappa^3 \gamma^{-1} =: G_f$$

*where $d = \max(n, m)$.*

## C   Results related to Algorithm SysId

In this subsection, a list of the properties related to Algorithm SysId is brought. More specifically:

- Theorem 3 gives the bounds on the estimated dynamics.

- Lemma 8 gives bounds on the state and the input while Algorithm SysId is running.

To this end, Some additional notations will be required along with the proofs. Let

- $J(\mathcal{A}|A, B, \{w\}, \{r\})$ be representing the total cost associated with executing the algorithm $\mathcal{A}$ over the $T$ time steps. With some abuse of notations, one can say $J(K|A, B, \{w\}, \{r\})$ shows the total cost associated with executing the linear controller $K$,

- $x_k(\mathcal{A}|A, B, \{w\}, \{r\})$ be the state visited at time $k$, and

- $u_k(\mathcal{A}|A, B, \{w\}, \{r\})$ be the control input at time $k$.

Also, if instead of $(A, B)$ in the above notations, $(\hat{A}, \hat{B})$ are used, it means that they are associated with the identified system instead of the actual one.

**Theorem 3 (Theorem 19 in (Hazan et al., 2020))** *If the system identification algorithm is run for $T_0$ steps, the output pair $(\hat{A}, \hat{B})$ satisfies, with probability $1 - \delta$, that $\|\hat{A} - A\|_{T_0}, \|\hat{B} - B\|_{T_0} \leq \epsilon_{A,B}$, where*

$$T_0 = 10^3 \lambda m n^2 \kappa^{10} \frac{\kappa_w^2}{\gamma^2 \epsilon_{A,B}^2} \log \frac{\kappa m n}{\delta}.$$

**Lemma 8** *Assume that Algorithm SysId is run for $T_0$ steps. Select the input as $u_k = K x_k + \eta_k$, where $\eta_k = [\eta_{k1}, ..., \eta_{km}]^T$, $\eta_{kj} \sim \{\pm 1\}$, $j = 1, ..., m$. Define*

$$D_{id} := \frac{\kappa^3}{\gamma}(\kappa_w + \kappa_b \sqrt{m}) + \sqrt{m}.$$

*Then,*

$$\|x_k\| \leq D_{id}, \ \|u_k\| \leq D_{id}, \tag{27}$$

$$J_0 = \sum_{k=1}^{T_0} \|c_k(x_k, u_k) \ - \min_{K \in \mathcal{K}} c_k(x_k, u_k)\| \leq 4 T_0 G_c D_{id}^2. \tag{28}$$

*Proof.* From the strong stability of $K$, for $x_k$, one has

$$\|x_{k+1}\| \leq \|\sum_{i=0}^{k}(A + BK)^{k-i}(w_i + B\eta_i)\|$$

$$\leq \sum_{i=0}^{k}\kappa^2(1 - \gamma)^{k-i}(\kappa_w + \kappa_b \|\eta_i\|).$$

Based on $\eta_{kj} \sim \{\pm 1\}^m$, one has $\|\eta_k\| \leq \sqrt{m}$. As a result, using the fact that $\sum_{i=0}^{k}(1 - \gamma)^i \leq \frac{1}{\gamma}$

$$\|x_k\| \leq \frac{\kappa^2}{\gamma}(\kappa_w + \kappa_b \sqrt{m}). \tag{29}$$

For $\|u_k\|$, one can derive the following

$$\|u_k\| \leq \|K x_k + \eta_k\| \leq \kappa \frac{\kappa^2}{\gamma}(\kappa_w + \kappa_b \sqrt{m}) + \sqrt{m} =: D_{id}.$$

Next, an upper bound for $\|c_k(x_k, u_k) - \min_{K \in \mathcal{K}} c_k(x_k, u_k)\|$ is computed. Based on Assumption 5, one has

$$\|c_k(x_k, u_k) - \min_{K \in \mathcal{K}} c_k(x_k, u_k)\|$$

$$\leq G_c D_{id} \|x_k - x_k(K^{\text{Opt}}|A, B, \{w\}, \{r\})\| + G_c D_{id} \|u_k - u_k(K^{\text{Opt}}|A, B, \{w\}, \{r\})\|$$

$$\leq 2 G_c D_{id}^2 + 2 G_c D_{id}^2 \leq 4 G_c D_{id}^2.$$

The result in equation 28 is concluded by summing the above inequality over $T_0$ steps.

## D    Result related to Algorithm 1

The following lemma provides an estimation of the upper bounds for the state, control input, and perturbation during the tracking stage.

**Lemma 9** *In the tracking step of Algorithm 1 for $k \geq T_0 + 1$ subsequently,*

$$\|x_k^\pi\| \leq \frac{\kappa^2}{\gamma}(\kappa_w + \kappa_b^2 \frac{\kappa^3}{\gamma}(\kappa_w + E_{w,T_0} + \frac{\kappa^2}{\gamma}(\kappa_w + \kappa_b\sqrt{m})) + \kappa_b^2 \frac{\kappa^3}{\gamma}\kappa_r) =: D_x, \tag{30}$$

$$\|u_k^\pi\| \leq \kappa D_x + \kappa_b \frac{\kappa^3}{\gamma}(\kappa_w + E_{w,T_0} + \frac{\kappa^2}{\gamma}(\kappa_w + \kappa_b\sqrt{m})) + \kappa_b \frac{\kappa^3}{\gamma}\kappa_r =: D_u. \tag{31}$$

*and*

$$\|\hat{w}_k - w_k\| \leq \epsilon_{A,B}(D_x + (\kappa D_x + \kappa^3 \frac{\kappa_b}{\gamma}(\kappa_w + E_{w,T_0} + \frac{\kappa^2}{\gamma}(\kappa_w + \kappa_b\sqrt{m})) + \kappa^3 \frac{\kappa_b}{\gamma}\kappa_r)) =: E_w, \tag{32}$$

$$\epsilon_{A,B}(\kappa D_{x,T_0} + \kappa_b \frac{\kappa^3}{\gamma}(\frac{\kappa^2}{\gamma}(\kappa_w + \kappa_b\sqrt{m})) + \kappa_b \frac{\kappa^3}{\gamma}\kappa_r) =: E_{w,T_0}$$

*where $D_x$, $D_u$ and $E_w$ are the upper bounds to the state, control input and perturbation estimation error, respectively. $E_{w,T_0}$ is the upper bound on the perturbation error at $k = T_0$.*

*Proof.* We prove the result by induction. First note that for equation 1 if the input is chosen as

$$u_k^\pi = Kx_k + \sum_{j=1}^{m_w} M^{[j-1]}\hat{w}_{k-j} + \sum_{s=0}^{m_r} P^{[s]}r_{k-s}, \tag{33}$$

the state $x_{k+1}^\pi$ reads

$$x_{k+1}^\pi = \sum_{i=0}^{k}(A + BK)^{k-i}(w_i + B\sum_{j=1}^{m_w} M^{[j-1]}\hat{w}_{k-j} + B\sum_{s=0}^{m_r} P^{[s]}r_{k-s}).$$

As a result,

$$\|x_{k+1}^\pi\| \leq \|\sum_{i=0}^{k}(A + BK)^{k-i}(w_i + B\sum_{j=1}^{m_w} M^{[j-1]}\hat{w}_{k-j} + B\sum_{s=0}^{m_r} P^{[s]}r_{k-s})\| \tag{34}$$

$$\leq \sum_{i=0}^{k}\kappa^2(1-\gamma)^{k-i}(\kappa_w + \kappa_b\|\hat{w}_k\|\sum_{j=1}^{m_w}\kappa_b\kappa^3(1-\gamma)^{t-1} + \kappa_b\|r_k\|\sum_{s=0}^{m_r}\kappa_b\kappa^3(1-\gamma)^s).$$

At time step $k = T_0$, one defines $\hat{w}_k = x_{T_0}$ whose upper bound is computed in equation 29. Thus, for $k = T_0$, one has

$$\|\hat{w}_{T_0}\| \leq \frac{\kappa^2}{\gamma}(\kappa_w + \kappa_b\sqrt{m}).$$

Based on the fact that $\sum_{i=0}^{k}(1-\gamma)^i \leq \frac{1}{\gamma}$ for $k = T_0$, one gets

$$\|x_{T_0+1}^\pi\| \leq \frac{\kappa^2}{\gamma}(\kappa_w + \kappa_b^2 \frac{\kappa^3}{\gamma}(\frac{\kappa^2}{\gamma}(\kappa_w + \kappa_b\sqrt{m})) + \kappa_b^2 \frac{\kappa^3}{\gamma}\kappa_r) =: D_{x,T_0+1}.$$

Then,

$$\|u_{T_0+1}^{\pi}\| \leq \|Kx_{k+1}^{\pi}\| + \sum_{j=1}^{m_w} \|M^{[j-1]}\hat{w}_{k-j}\| + \sum_{s=0}^{m_r} \|P^{[s]}r_{k-s}\|$$

$$\leq \kappa D_{x,T_0+1} + \|\hat{w}_k\| \sum_{j=1}^{m_w} \kappa_b \kappa^3 (1-\gamma)^{t-1} + \|r_k\| \sum_{s=0}^{m_r} \kappa_b \kappa^3 (1-\gamma)^s$$

$$\leq \kappa D_{x,T_0+1} + \kappa_b \frac{\kappa^3}{\gamma} (\frac{\kappa^2}{\gamma}(\kappa_w + \kappa_b\sqrt{m})) + \kappa_b \frac{\kappa^3}{\gamma}\kappa_r := D_{u,T_0+1}.$$

Assuming that $\|A - \hat{A}\|, \|B - \hat{B}\| \leq \epsilon_{A,B}$, at $k > T_0$

$$\|\hat{w}_k - w_k\| \leq \|((A - \hat{A})x_k + (B - \hat{B})u_k)\| \leq \epsilon_{A,B}\|x_k\| + \epsilon_{A,B}\|u_k\|.$$

Thus,

$$\|\hat{w}_k - w_k\| \leq \epsilon_{A,B}(\frac{\kappa^2}{\gamma}(\kappa_w + \kappa_b^2 \frac{\kappa^3}{\gamma}(\frac{\kappa^2}{\gamma}(\kappa_w + \kappa_b\sqrt{m}) + \kappa_b^2 \frac{\kappa^3}{\gamma}\kappa_r)) +$$

$$\epsilon_{A,B}(\kappa D_{x,T_0} + \kappa_b \frac{\kappa^3}{\gamma}(\frac{\kappa^2}{\gamma}(\kappa_w + \kappa_b\sqrt{m})) + \kappa_b \frac{\kappa^3}{\gamma}\kappa_r) := E_{w,T_0},$$

and

$$\|\hat{w}_k\| \leq \max(E_{w,T_0} + \kappa_w, \|\hat{w}_{T_0}\|).$$

Then, if the upper bound for $\|\hat{w}_k\|$ is replaced with $E_{w,T_0} + \kappa_w + \|\hat{w}_{T_0}\|$ in equation 34, the bounds in equation 30-equation 31 are concluded.

## E   Helper results for the proof of the regret bound

To prove the regret bound, a few results are needed.

- Lemma 10 is a technical result to be used in Lemma 11.

- Lemma 11 gives an upper bound for the difference in the costs for the real and estimated systems using a linear controller.

- Lemma 12 provides an upper bound for the difference between the cost of using the tracking algorithm and the minimum cost that can be achieved by the same class of controller.

**Lemma 10** *For any matrix pair $P, \Delta P$, such that $\|P\|, \|P + \Delta P\| \leq 1 - \gamma$, it holds*

$$\sum_{i=0}^{\infty} \|(P + \Delta P)^i - P^i\| \leq \frac{\|\Delta P\|}{\gamma^2}.$$

*Proof.* This proof is based on an inductive argument. First, we prove that the inequality $\|(P+\Delta P)^i - P^i\| \leq \|\Delta P\|i(1-\gamma)^{i-1}$ holds true. Then, the validity of this claim can be easily verified for $i = 0$ and $i = 1$. Next, it is shown that this claim is valid for the case $i + 1$. Observe that

$$\|(P + \Delta P)^{i+1} - P^{i+1}\| \leq \|(P + \Delta P)(P + \Delta P)^i - (P)(P)^i\|$$

$$\leq \|P((P + \Delta P)^i - P^i) + \Delta P(P + \Delta P)^i\|$$

$$\leq \|P((P + \Delta P)^i - P^i)\| + \|\Delta P(P + \Delta P)^i\|$$

It is known from the claim that $\|(P+\Delta P)^i - P^i\| \leq i\|\Delta P\|(1-\gamma)^{i-1}$ and it is assumed that $\|P+\Delta P\| \leq 1-\gamma$, thus,

$$
\begin{aligned}
\|P((P+\Delta P)^i - P^i)\| + \|\Delta P(P+\Delta P)^i\| &\leq \|P\|\|((P+\Delta P)^i - P^i)\| + \|\Delta P(P+\Delta P)^i\| \\
&\leq (1-\gamma)i\|\Delta P\|(1-\gamma)^{i-1} + \|\Delta P\|(1-\gamma)^i \\
&\leq (i+1)(1-\gamma)^i\|\Delta P\|.
\end{aligned}
$$

Then,

$$
\begin{aligned}
\sum_{i=0}^{\infty} \|(P+\Delta P)^i - P^i\| = \sum_{i=-1}^{\infty} \|(P+\Delta P)^{(i+1)} - P^{(i+1)}\| &\leq \\
\sum_{i=-1}^{\infty} (i+1)(1-\gamma)^i\|\Delta P\| &= \sum_{i=0}^{\infty} (i)(1-\gamma)^{i-1}\|\Delta P\|.
\end{aligned}
$$

(Hazan et al., 2020) in Lemma 17 showed that $\sum_{i=0}^{\infty} i(1-\gamma)^{i-1} \leq \frac{1}{\gamma^2}$. Thus, the proof is concluded.

**Lemma 11** *Assuming that* $\|A - \hat{A}\| \leq \epsilon_{A,B}, \|B - \hat{B}\| \leq \epsilon_{A,B}$, *where* $\epsilon_{A,B} \leq 0.25k^{-3}\gamma$, *and that* $K$ *is* $(\kappa, \gamma)$-*strongly stable with respect to the pair* $(A, B)$. *Then, from Lemma 9, for any perturbation sequence satisfying* $\|w_k - \hat{w}_k\| \leq E_w$, *and it is assumed that* $\|\hat{w}_0\| \leq W_0$, *the following statement holds*

$$
|J(K|\hat{A}, \hat{B}, \{\hat{w}\}, \{r\}) - J(K|A, B, \{w\}, \{r\})| \leq poly(\kappa, \frac{1}{\gamma}, \lambda, m, n, \kappa_w, \kappa_z)G_c T T_0^{-1/2}.
$$

*Proof.* One has $\|L\| \leq 1 - \gamma$ for $(A, B)$ and $\|\hat{L}\| \leq 1 - \gamma + 2\kappa^3\epsilon_{A,B}$ from Lemma 1. It can be said that $\hat{L} = L + \Delta L$ and $\|\Delta L\| \leq 2\kappa^3\epsilon_{A,B}$. Using Lemma 10 for $L$ and $\hat{L}$ it can be stated that if one take $\epsilon_{A,B} = \frac{\gamma}{4\kappa^3}$, one will have $\|L\|$ and $\|\hat{L}\| \leq 1 - \frac{\gamma}{2}$. The linear controller K is as equation 3, and is $(\kappa, \gamma)$-strongly stable for (A, B). It holds

$$
x_k(K|A, B, \{w\}, \{r\}) = \sum_{i=0}^{k} (A + Bk_{fb})^i(BK_{ff}z_{k-i} + w_{k-i}).
$$

Thus, with the assumption that $\kappa_b \leq \kappa$ without any loss of generality,

$$
\|x_k(K|A, B, \{w\}, \{r\})\| \leq \frac{\kappa^4\kappa_z}{\gamma} + \frac{\kappa^2\kappa_w}{\gamma}.
$$

Similarly it can be said that by Lemma 1, since $\epsilon_{A,B} = \frac{\gamma}{4\kappa^3}$, $\kappa > 1$, and $0 \leq \gamma \leq 1$ one has $\|\hat{A}\| \leq \|A\| + \epsilon_{A,B}$ *and* $\|\hat{B}\| \leq \|B\| + \epsilon_{A,B}$, hence

$$
\begin{aligned}
\|\hat{A}\| &\leq \kappa + \frac{\gamma}{4\kappa^3} \leq \kappa + \frac{1}{4\kappa^3} \leq 2\kappa, \\
\|\hat{B}\| &\leq \kappa + \frac{\gamma}{4\kappa^3} \leq \kappa + \frac{1}{4\kappa^3} \leq 2\kappa.
\end{aligned}
$$

Thus, K is $(2\kappa, \frac{\gamma}{2})$-strongly stable for $(\hat{A}, \hat{B})$ and

$$
\|x_k(K|\hat{A}, \hat{B}, \{\hat{w}\}, \{r\})\| \leq \frac{(2\kappa)^4\kappa_z}{\gamma/2} + \frac{(2\kappa)^2\kappa_z}{\gamma/2}W_0 \leq \frac{32\kappa^4\kappa_z}{\gamma} + \frac{8\kappa^2W_0}{\gamma}.
$$

Subsequently:

$$\|x_{k+1}(K|A, B, \{w\}, \{r\}) - x_{k+1}(K|\hat{A}, \hat{B}, \{\hat{w}\}, \{r\})\|$$

$$\leq \sum_{i=0}^{k} \|(A + BK_{fb})^i w_{k-i} - (\hat{A} + \hat{B}k_{fb})^i \hat{w}_{k-i} + (A + BK_{fb})^i BK_{ff}z_{k-i} - (\hat{A} + \hat{B}K_{fb})^i \hat{B}K_{ff}z_{k-i}\|$$

$$\leq \sum_{i=0}^{k} (\|(A + BK_{fb})^i w_{k-i} - (A + BK_{fb})^i \hat{w}_{k-i}\| + \|(A + BK_{fb})^i \hat{w}_{k-i} - (\hat{A} + \hat{B}k_{fb})^i \hat{w}_{k-i}\|$$

$$+ \|(A + BK_{fb})^i BK_{ff}z_{k-i} - (\hat{A} + \hat{B}K_{fb})^i \hat{B}K_{ff}z_{k-i}\|).$$

Starting from the first term, one has

$$\sum_{i=0}^{k} \|(A + BK_{fb})^i w_{k-i} - (A + Bk_{fb})^i \hat{w}_{k-i}\|) \leq \frac{\kappa^2 E_w}{\gamma} + \kappa^2 (1 - \gamma)^k W_0.$$

For the second term, one has $\|\hat{A} + \hat{B}k_{fb}\| \leq Q^{-1}\hat{L}Q$ and $\|A + Bk_{fb}\| \leq Q^{-1}LQ$. Since $\|L\|, \|\hat{L}\| \leq 1 - \frac{\gamma}{2}$, then $\|\hat{A} + \hat{B}k_{fb}\|, \|A + Bk_{fb}\| \leq Q^{-1}(1 - \frac{\gamma}{2})Q$. Additionally, $\|Q\|, \|Q^{-1}\| \leq \kappa$. Using Lemma 10 and knowing $\|\Delta L\| \leq 2\kappa^3 \epsilon_{A,B}$,

$$\sum_{i=0}^{k} (\|(A + BK_{fb})^i \hat{w}_{k-i} - (\hat{A} + \hat{B}k_{fb})^i \hat{w}_{k-i}\|)$$

$$\leq \sum_{i=0}^{k} (\|(Q^{-1}LQ)^i \hat{w}_{k-i} - (Q^{-1}\hat{L}Q)^i \hat{w}_{k-i}\|)$$

$$\leq \kappa^2 W_0 \sum_{i=0}^{k} \|(L)^i - (\hat{L})^i\|$$

$$\leq 8\kappa^5 \frac{\epsilon_{A,B}}{\gamma^2} W_0.$$

For the third term, following the same steps from the last part,

$$\sum_{i=0}^{k} \|((A + BK_{fb})^i BK_{ff}z_{k-i} - (\hat{A} + \hat{B}K_{fb})^i \hat{B}K_{ff}z_{k-i})\|$$

$$\leq \frac{4\kappa^4 (2\kappa^3 \epsilon_{A,B}\kappa_z)}{\gamma^2}.$$

Thus,

$$\|x_{k+1}(K|A, B, \{w\}, \{r\}) - x_{k+1}(K|\hat{A}, \hat{B}, \{\hat{w}\}, \{r\})\|$$

$$\leq \frac{\kappa^2 E_w}{\gamma} + \kappa^2 (1 - \gamma)^k W_0 + \frac{8W_0 \kappa^5 \epsilon_{A,B}}{\gamma^2} + \frac{8\kappa^7 \epsilon_{A,B}\kappa_z}{\gamma^2}.$$

From assumption 5, it can be shown that $\|x_k\|, \|u_k\| \leq D$ and $\|\nabla_x c_k(x, u)\|, \|\nabla_u c_k(x, u)\| \leq G_c D$. With abuse of the use of the notation, let $\hat{D}$ denote the bound related to the identified system. It holds:

$$u_{k+1}(K|\hat{A}, \hat{B}, \{\hat{w}\}, \{r\}) = K_{fb}x_{k+1}(K|\hat{A}, \hat{B}, \{\hat{w}\}) + K_{ff}z_{k+1}.$$

Thus,

$$\|u_{k+1}(K|\hat{A}, \hat{B}, \{\hat{w}\}, \{r\})\| \leq \frac{8\kappa^3 W_0}{\gamma} + \frac{32\kappa^5 \kappa_z}{\gamma} + \kappa\kappa_z =: \hat{D}.$$

As a result, we have

$$|J(K|\hat{A}, \hat{B}, \{\hat{w}\}, \{r\}) - J(K|A, B, \{w\}, \{r\})| \leq TG_c\hat{D}\|x_{k+1}(K|A, B, \{w\}, \{r\}) - x_{k+1}(K|\hat{A}, \hat{B}, \{\hat{w}\}, \{r\})\|$$
$$\leq TG_c\left(\frac{8\kappa^3 W_0}{\gamma} + \frac{32\kappa^5 \kappa_z}{\gamma} + \kappa\kappa_z\right)\left(\frac{8\kappa^5 W_0\epsilon_{A,B}}{\gamma^2} + \frac{8\kappa^7 \epsilon_{A,B}\kappa_z}{\gamma^2} + \kappa^2(1-\gamma)^k W_0 + \frac{\kappa^2 E_w}{\gamma}\right).$$

From equation 32, one can derive $E_w \leq \text{poly}(\kappa, \kappa_b, m, \kappa_w, \gamma^{-1}, \kappa_r)\epsilon_{A,B}$, and $W_0 \leq E_w$. Also, from Theorem 3, it is known that $T_0 = \epsilon_{A,B}^{-2}\text{poly}(\kappa, \kappa_w, m, n)$. Thus, one can write the above inequality as

$$|J(K|\hat{A}, \hat{B}, \{\hat{w}\}, \{r\}) - J(K|A, B, \{w\}, \{r\})| \leq \text{poly}(\kappa, \frac{1}{\gamma}, \lambda, m, n, \kappa_w, \kappa_z)G_c T\, T_0^{-1/2}.$$

**Lemma 12** *Let Assumptions 1-5 hold. Let $x_k^{\pi^*}$ denote the state using the optimal memory-augmented controller $u_k^{\pi^*}$. Set $H = m_w = m_r$. Let $Y_{H,k}^* := [M^*, .., M^*, P^*, ..., P^*]$ denote the optimal weights learned by Algorithm 1, each one of the weights repeated for $H$ times, and $\tilde{x}_k^\pi(Y_{H,k}^*)$, $\tilde{u}_k^\pi(Y_{H,k}^*)$ denote the truncated states and control using these optimal weights according to equation 13-equation 14. Then*

$$\left|c_k(\tilde{x}_k^\pi(Y_{H,k}^*) - r_k, \tilde{u}_k^\pi(Y_{H,k}^*)) - c_k\left(x_k^{\pi^*} - r_k, u_k^{\pi^*}\right)\right| \leq 2\,G_c D^2 \kappa^3 (1-\gamma)^H.$$

*Proof of Lemma 12:* Stacking optimal learned weights for k times makes $Y_{0:k}^* := [M^*, ..., M^*, P^*, ..., P^*]$, and then stacking them for $H$ times defines $Y_{H,k}^* := [M^*, ..., M^*, P^*, ..., P^*]$. Based on Lemma 4, one has

$$\left|c_k(\tilde{x}_k^\pi(Y^*) - r_k, \tilde{u}_k^\pi(Y^*)) - c_k\left(x_k^{\pi^*} - r_k, u_k^{\pi^*}\right)\right| \leq G_c\,D\,\|(x_k^K(Y_{0:k-1}^*) - r_k) - (\tilde{x}_k^\pi(Y_{H,k}^*) - r_k)\|$$
$$+ G_c D\,\|u_k^K(Y_{0:k-1}^*) - \tilde{u}_k^\pi(Y_{H,k}^*)\| \leq 2G_c D^2 \kappa^3 (1-\gamma)^H.$$

This completes the proof.

**Theorem 4** *Suppose $\mathcal{A} :=$ Algorithm RobTrack is executed under Assumptions 1-5. Let $H = m_w = m_r$. Select the learning rate $\eta$ and the memory size $H$ to satisfy $\eta = \mathcal{O}(\frac{1}{G_c\kappa_w\sqrt{T}})$, $H = \mathcal{O}(\log \frac{\kappa^2 T}{\gamma})$, and $T_0 = T^{2/3}$. The regret of Algorithm RobTrack on the identified system $(\hat{A}, \hat{B})$ and the perturbation $\{\hat{w}\}$ is*

$$J(\mathcal{A}|\hat{A}, \hat{B}, \{\hat{w}\}, \{r\}) - J(\mathcal{A}^{opt}|\hat{A}, \hat{B}, \{\hat{w}\}, \{r\}) = \mathcal{O}(\sqrt{T}),$$

*where $J(\mathcal{A}^{\text{opt}}|\hat{A}, \hat{B}, \{\hat{w}\}, \{r\})$ denotes the total cost associated with the optimal memory-augmented policy $u_k^{\pi^*}$.*

*Proof:* To begin, one has

$$\begin{aligned}
J(\mathcal{A}|A,B,\{w\},\{r\}) - J(\mathcal{A}^{\mathrm{opt}}|A,B,\{w\},\{r\}) &= \sum_{k=1}^{T} c_k(e_k,u_k) - \sum_{k=1}^{T} c_k\big(x_k^{\pi^*} - r_k, u_k^{\pi^*}\big) \\
&= \underbrace{\sum_{k=1}^{T} c_k(e_k(Y_{0:k-1}), u_k(Y_{0:k-1})) - \sum_{k=1}^{T} f_k(Y_{H,k})}_{\alpha_T} \\
&\quad + \underbrace{\sum_{k=1}^{T} f_k(Y_{H,k}) - \sum_{k=1}^{T} f_k(Y^*)}_{\beta_T} \\
&\quad + \underbrace{\sum_{k=1}^{T} f_k(Y^*) - \sum_{k=1}^{T} c_k\big(x_k^{\pi^*} - r_k, u_k^{\pi^*}\big)}_{\zeta_T},
\end{aligned}$$

where $Y^* = [M^{[0]*}, ..., M^{[H-1]*}, P^{[0]*}, ..., P^{[H-1]*}] \in (\mathbb{R}^{m \times n})^{2H}$ denote the optimal weights learned by Algorithm 1, satisfying the conditions in 3.

The regret analysis is split into three parts: $\alpha_T$ denotes the difference between the cost of Algorithm 1 and the truncated cost. $\beta_T$ denotes the difference between the truncated and optimal truncated costs. $\zeta_T$ denotes the difference between the optimal truncated cost and the optimal memory-augmented control policy.

The bound of the first term $\alpha_T$ is given by

$$\begin{aligned}
|c_k(e_k,u_k) - f_k(Y_{H,k})| &\le G_c\, D \, \|(x_k^K(Y_{0:k-1}) - r_k) - (\tilde{x}_k^\pi(Y_{H,k}) - r_k)\| + G_c D \, \|u_k^K(Y_{0:k-1}) - \tilde{u}_k^\pi(Y_{H,k})\| \\
&\le 2 G_c D^2 \kappa^3 (1-\gamma)^H,
\end{aligned}$$

where one can use Lemma 4 to get the above result. Therefore,

$$\|\alpha_T\| = \|\sum_{k=1}^{T} c_k(e_k,u_k) - \sum_{k=1}^{T} f_k(Y_{H,k})\| \le 2T\, G_c D^2 \kappa^3 (1-\gamma)^H = \mathcal{O}(\sqrt{T}), \tag{35}$$

where the last equality is obtained based on $H = \mathcal{O}(log\, T)$.

The term $\beta_T$ can be bounded by Theorem 4.6 of Agarwal et al. (2019) and the results of Lemmas 6 and 7 as

$$\sum_{k=1}^{T} f_k(Y_{H,k}) - \sum_{k=1}^{T} f_k(Y^*) \le \frac{1}{\eta} M_b^2 + T G_f^2 \eta + L_f H^2 \eta G_f T,$$

where $M_b := 2\sqrt{d}\kappa_b \kappa^3 \gamma^{-1}$, $d = \max(n,m)$. By selecting $\eta = \mathcal{O}(\frac{1}{G_c \kappa_w \sqrt{T}})$, $H = \mathcal{O}(log\,(T))$, then $\beta_T = \mathcal{O}(\sqrt{T})$.

The last term is the difference between the truncated cost of the algorithm and the cost by the optimal memory-augmented controller. For the third term, using Lemma 12,

$$\sum_{k=1}^{T} f_k(Y_{H,k}^*) - \sum_{k=1}^{T} c_k\big(x_k^{\pi^*} - r_k, u_k^{\pi^*}\big) \le 2T\, G_c D^2 \kappa^3 (1-\gamma)^H = \mathcal{O}(\sqrt{T}),$$

where the last equality is obtained based on $H = \mathcal{O}(log\,(T))$. Observe that

- If $\epsilon_{A,B} \le \frac{\gamma}{4\kappa^3}$, Lemma 1 guarantees that $k$ is $(2\kappa, \frac{\gamma}{2})$-strongly stable on $(\hat{A}, \hat{B})$,

- If $\epsilon_{A,B} \le \mathrm{poly}(\kappa, \frac{1}{\gamma})$, the iterates obtained by running Algorithm $\mathcal{A}$ (trajectory tracking algorithm) satisfy $\|x_k\|, \|\hat{w}_k\| \le \mathrm{poly}(\kappa, \frac{1}{\gamma}, n)(1 + \kappa_w)$, as guaranteed by Lemma 9.

Given the aforementioned observations and the proof before that, it is ensured that

$$J(\mathcal{A}|\hat{A}, \hat{B}, \{\hat{w}\}, \{r\}) - J(\mathcal{A}^{\mathrm{opt}}|\hat{A}, \hat{B}, \{\hat{w}\}, \{r\}) \leq \mathcal{O}(\sqrt{T}).$$

Also, for the sake of completeness and self-containment of this article, it can be mentioned that from (Yaghmaie & Modares, 2023), it is known that

$$J(\mathcal{A}|A, B, \{w\}, \{r\}) - J(K|A, B, \{w\}, \{r\}) \leq \mathcal{O}(\sqrt{T}),$$

and following the same steps as before one can conclude

$$J(\mathcal{A}|\hat{A}, \hat{B}, \{\hat{w}\}, \{r\}) - J(K|\hat{A}, \hat{B}, \{\hat{w}\}, \{r\}) \leq \mathcal{O}(\sqrt{T}).$$

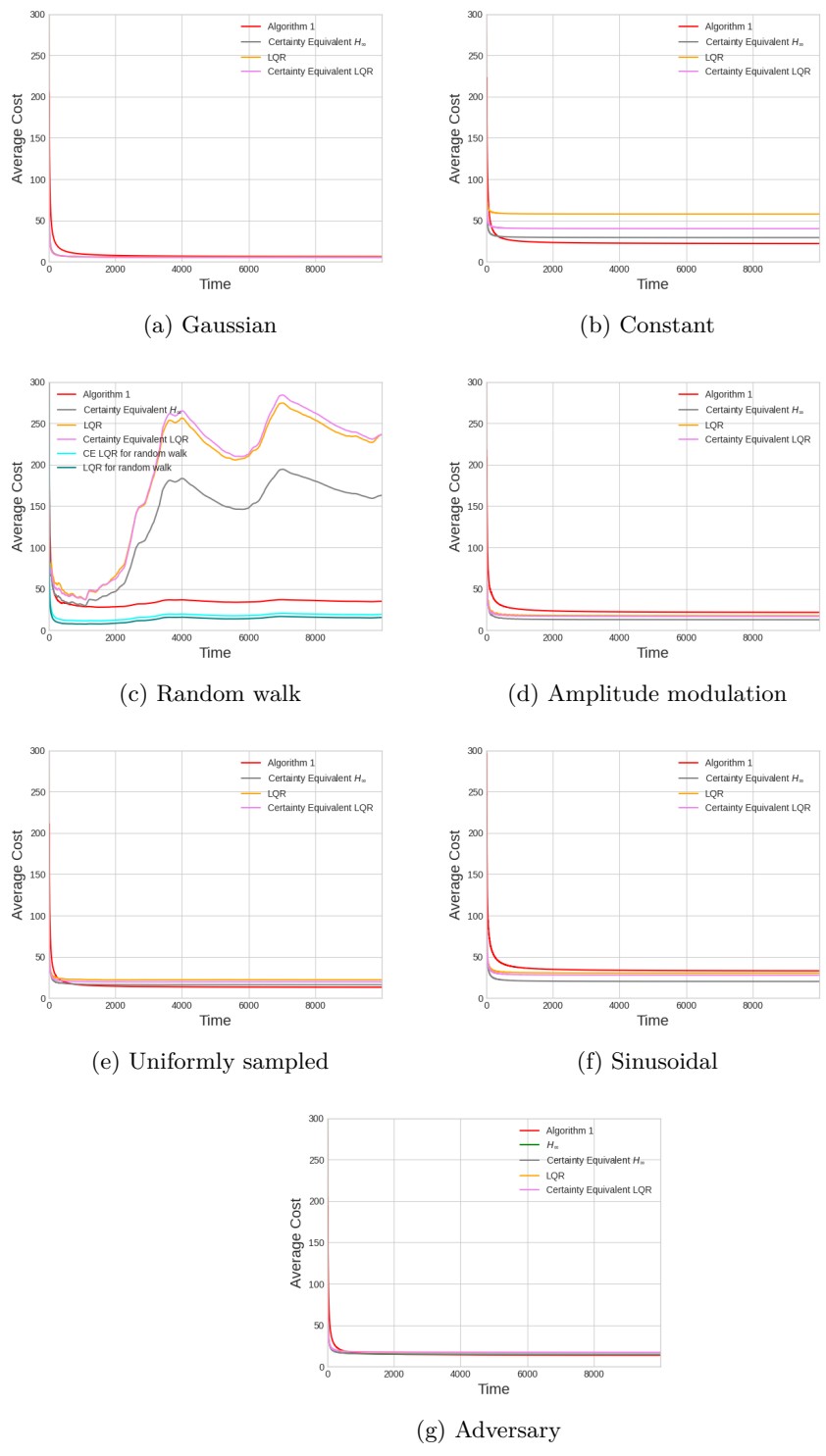

Figure 9: The changes in the average cost $\bar{J}_T$, as given by equation 6 over time $T$ is compared between Algorithm 1 and other control methods, namely certainty equivalent $H_\infty$-control, certainty equivalent LQR control, and LQR control with knowledge of system dynamics, for different types of disturbances such as Gaussian, random walk, uniformly sampled, constant, amplitude modulation, sinusoidal, and adversary disturbances.

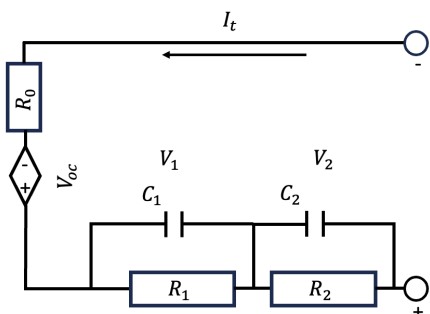

Figure 10: Circuit of a Lithium-ion battery.

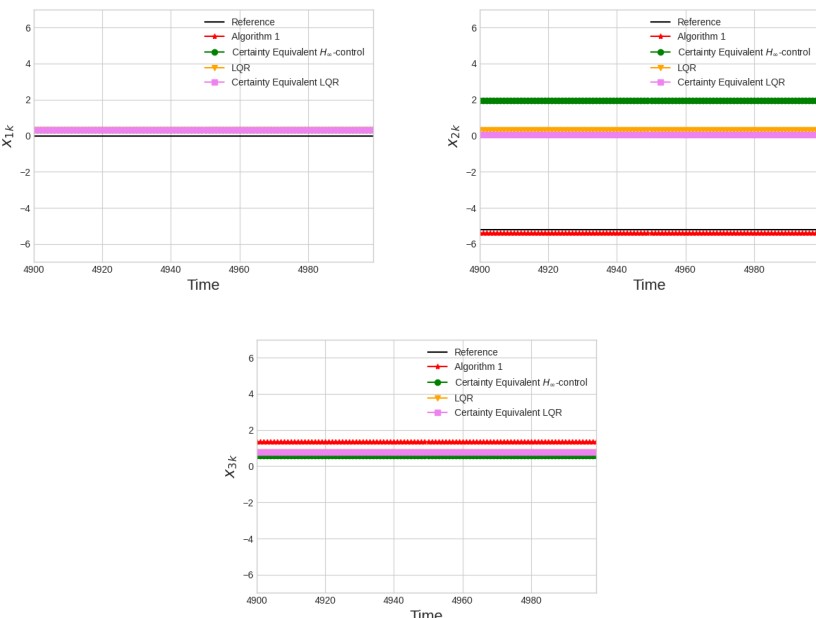

Figure 11: Tracking performance for constant disturbance for the presented Algorithm 1, versus certainty equivalent $H_\infty$-control, certainty equivalent LQR control, and LQR control knowing the dynamics of the system.

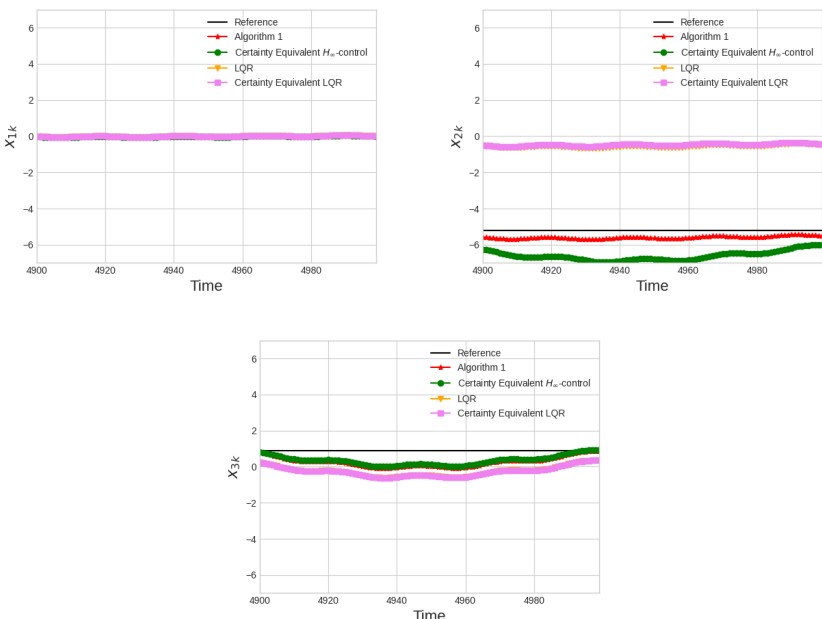

Figure 12: Tracking performance for amplitude modulation disturbance for the presented Algorithm 1, versus certainty equivalent $H_\infty$-control, certainty equivalent LQR control, and LQR control knowing the dynamics of the system.

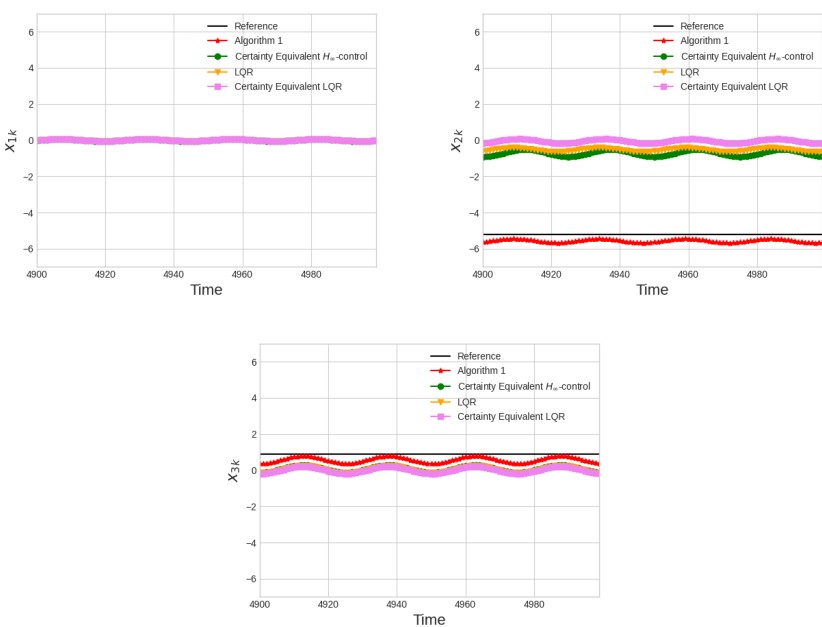

Figure 13: Tracking performance for sinusoidal disturbance for the presented Algorithm 1, versus certainty equivalent $H_\infty$-control, certainty equivalent LQR control, and LQR control knowing the dynamics of the system.

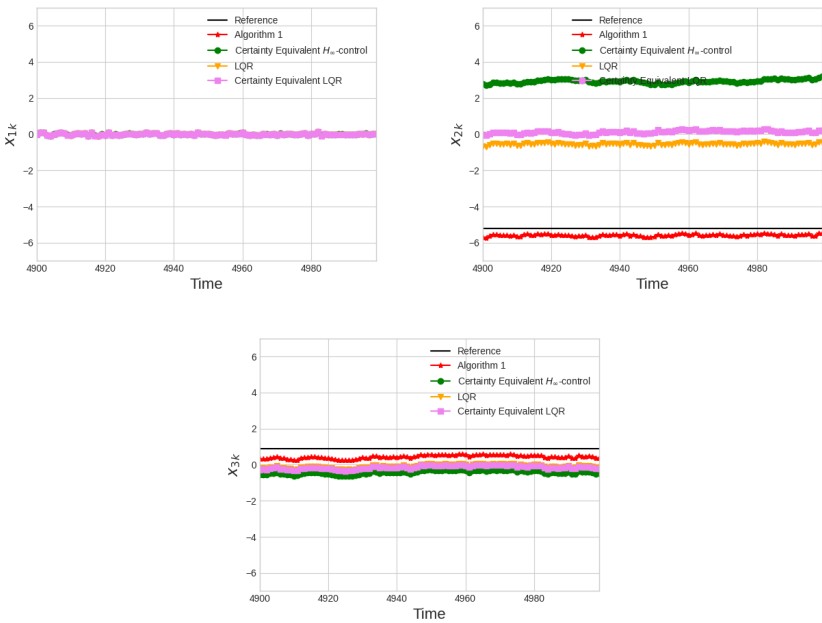

Figure 14: Tracking performance for Gaussian disturbance for the presented Algorithm 1, versus the certainty equivalent $H_\infty$-control, certainty equivalent LQR control, and LQR control knowing the dynamics of the system.

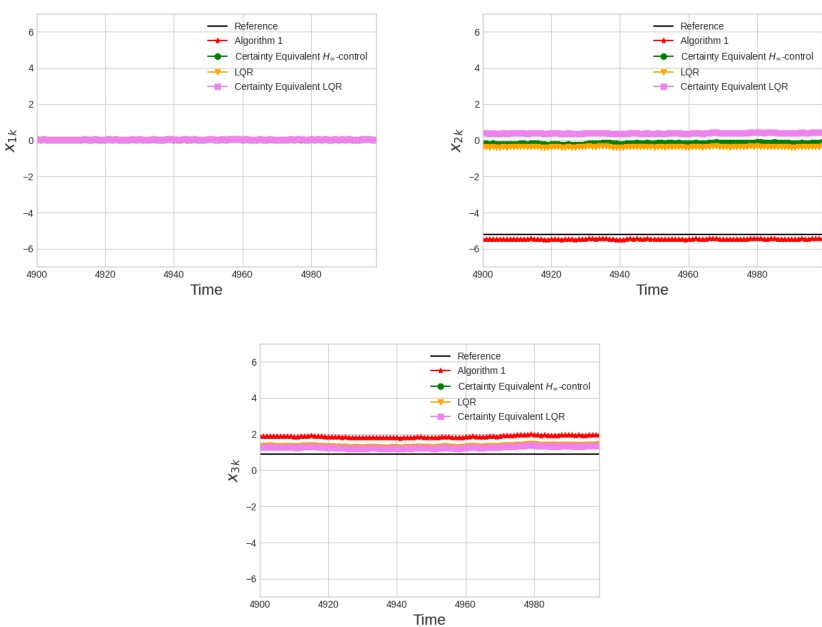

Figure 15: Tracking performance for the uniformly sampled disturbance for the presented Algorithm 1, versus certainty equivalent $H_\infty$-control, certainty equivalent LQR control, and the LQR control knowing the dynamics of the system.

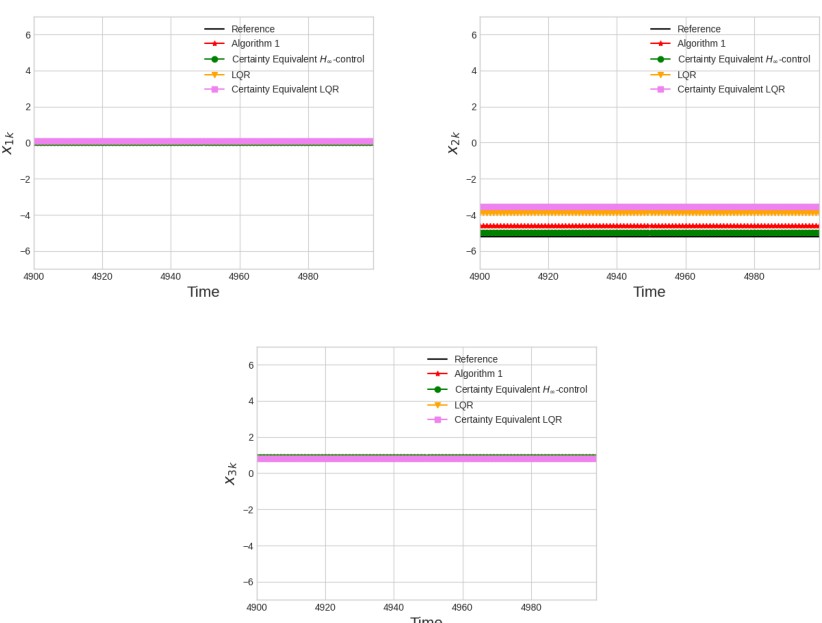

Figure 16: Tracking performance for the adversary disturbance for the presented Algorithm 1, versus the $H_\infty$-control, certainty equivalent $H_\infty$-control, certainty equivalent LQR control, and LQR control knowing the dynamics of the system.

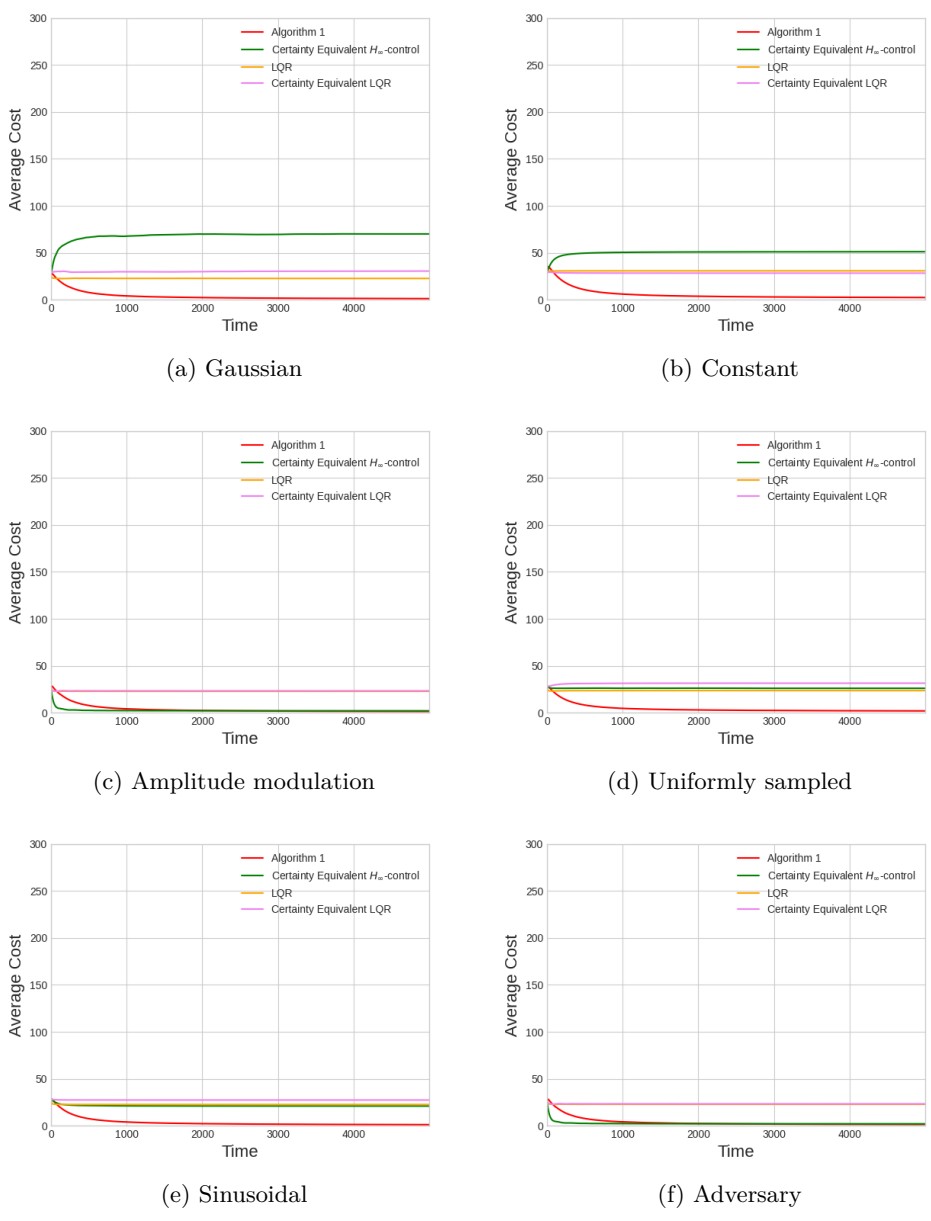

Figure 17: The changes in the average cost $\bar{J}_T$, as given by equation 6, over varying values of $T$, is compared between Algorithm 1 and other control methods, namely certainty equivalent $H_\infty$-control, certainty equivalent LQR control, and LQR control with knowledge of system dynamics, for different types of disturbances such as Gaussian, uniformly sampled, constant, amplitude modulation, sinusoidal, and adversary disturbances.

