# OpenReview forum: "Online Reference Tracking For Linear Systems with Unknown Dynamics and Unknown Disturbances"
_TMLR — Accepted by TMLR_

### Review · Reviewer_Mjcu · 2023-08-25

**Summary Of Contributions:**

The paper studies a method for online reference tracking for a system with unknown dynamics and disturbances. The first few sections of the paper provide preliminary and background information on tracking problems, building up to the main problem formulation of the paper in Problem 1. The proposed method in Algorithm 1 has two steps: 1) system identification by sending binary inputs into the system, followed by 2) tracking the error approximations and solving the state-tracking problem online.  Theorem 2 analyzes the regret of this method and Section 5 experimentally evaluates the method on the system in eq (22) under various disturbances.

**Audience:**

Yes

**Claims And Evidence:**

Yes

**Requested Changes:**

I do not have any

**Strengths And Weaknesses:**

Strengths
+ For the most part, the main problem formulation and algorithmic contribution are clearly motivated and stated.
  + I did get somewhat lost trying to understand Section 3.3, which seems to heavily rely on Yaghmaie & Modares (2023).
  + Problem 1 could be easier to understand with one more level of notation/equations. The last sentence says "design a memory-augmented control policy in the form of eq 9 to optimize eq 7" --- it could be nice to also write out the min/argmin equation to make it extremely concrete and easy to see the optimization problem of interest.
+ The experimental evaluation is clearly described and the baselines of C.E. LQR, and C.E. $H_\infty$ control are reasonable. The proposed methodology and this experiment seem to back up the claims and evidence, which is the acceptance criteria for TMLR.

Weaknesses
+ As stated in the introduction, the paper reads as a relatively straightforward combination of the optimal tracking of [Yaghmaie and Modares](https://openreview.net/forum?id=5nVJlKgmxp) with the non-stochastic control problem in [Hazan et al.](https://arxiv.org/abs/1911.12178).  Algorithm 1 of the submitted paper closely follows the structure of [Algorithm 1 in Hazan et al.](https://arxiv.org/abs/1911.12178) of performing system identification with random inputs and then solving the online control problem. The difference is that Hazan et. al don't perform reference tracking and consider a more general control cost while this paper uses the reference tracking using [Algorithm 1 in Yaghmaie and Modares](https://openreview.net/forum?id=5nVJlKgmxp). Experimentally, I do not completely understand the difficulty/novelty in combining these approaches, but the paper seems like it would be a good reference for anybody wanting to do this.
+ Even though the system identification portion using random perturbations was taken from Hazan et al., it seems important to justify or briefly say when that works. Especially if the approach is generalized to non-linear systems, there are many systems where random perturbations using control values of -1 and 1 would not be able to adequately recover the dynamics. Maybe this is ok for linear systems and there are results explaining why perturbations using control values of just -1 and 1 work and how fast that can approach the true system.
+ The formulation of the paper throughout the introduction and in problem 1 state that the method is robust to adversarial perturbations. However, experimentally, adversarial disturbances are not investigated --- Section 5.2 only lists uniformly sampled, constant, amplitude modulation, sinusoidal, Gaussian, and random walk disturbances. While these go beyond the usual Gaussian perturbance assumptions, they still don't involve an adversary. I have the same comment about the settings from [Yaghmaie and Modares](https://openreview.net/forum?id=5nVJlKgmxp) which also do not experimentally consider adversarial disturbances.
+ The experimental evaluation in setting 5.1 is for a very specific 2D linear dynamical system with very specific values set for the reference system. While it is a reasonable demonstration of the setting, I would be much more convinced by a system and reference tracking problem directly taken from a real-world dynamical system and application. The would be the most interesting for real non-linear systems (e.g., a vehicular, aerospace, or other robotic/mechanical system), so I acknowledge that it's out-of-scope for the submitted paper's contribution on linear systems.

Minor
+ In `Algorithm SysId`, $\mathbb{K}$ is used as the linear controller, but it should just be $K$ from definition 1? The other parts of the SysId algorithm block are also not very well-described in comparison to Algorithm 1 which briefly describes the inputs/initialization with text

---

> ### Author Response · Authors · 2023-11-21
>
> We appreciate your encouraging feedback on our manuscript.  We have conducted a thorough revision of the entire paper, taking into account the found issues and integrating feedback from the reviewers. These revisions have been systematically highlighted in blue throughout the document for clarity. Additionally, we have included a real-world experiment focused on tracking the states of Lithium-ion batteries, now presented in the Appendix for a comprehensive algorithm analysis. Your time and insightful input into our paper are sincerely valued. We hope we were able to answer all your concerns. Please refer to our detailed point-to-point response for further clarification on your comments.
>
> {Strengths:}
>
> 1. Thank you for acknowledging the clarity and motivation behind the main problem formulation and algorithmic contribution. We worked hard on this paper.
>
> 2. Section 3.3 introduces the policy class with the learnable parameters $M,\:P$ and discusses that the state upon execution of a policy from this class is linear in $M,\:P$. We have reorganized this section and brought the result as a lemma to enhance readability.
>
> 3. Thanks. Done. The equations for the policy and the optimization problem cost function are now brought in Problem 1.
>
> 4. Thank you for your positive feedback on the experimental evaluation and the reasonable choice of baselines, such as C.E. LQR and C.E. control.
>
> {Weaknesses:}
>
> 1. We greatly appreciate your positive feedback. While we understand that the paper may seem a bit unclear, our intention was to offer a comprehensive analysis of regret bounds, particularly in cases where the system dynamics are unknown. The goal was to elevate the tracking algorithm, inspired by Yaghmaie and Modares, to a fully data-driven approach, enhancing its practical relevance. If you have any more specific suggestions or questions, we would be delighted to address them. In the meantime, we will try to answer the questions you have so far from us.
>
> 2. In the newly added Remark 2, we tried to address the reason binary signal identification can be useful only for linear systems. In (Ljung, L. (1995). System identification toolbox: User's guide. Natick, MA, USA: MathWorks Incorporated.) in section 13.3 it is discussed that binary inputs are rich enough for the identification of linear systems.
>
> 3. We tried implementing the adversarial disturbance by adding the worst-case disturbance to our experiments. This disturbance can be formulated as a zero-sum problem. We added how this worst-case disturbance can be computed based on the simulation section of the paper. As we can see in the results, our control algorithm outperforms certainty equivalent $H_{\infty}$ in this setting. We believe this is because the computed gain based on the identified model in certainty equivalent methods might not be exactly the same gain that model-based $H_{\infty}$ yields.
>
> 4. We have now gone through the simulation section and revised the material there, especially the graphs and the explanation of the algorithms. Additionally, we included a new real-world experiment focusing on tracking Lithium-ion battery states. This has been added to the Appendix to present a complete analysis and assessment of the algorithm.
>
> {Minor:}
>
> Yes, you are right. We have now fixed the typo that we had in the algorithms. Also, we have now added a brief explanation of how the SysId algorithm works along the text.

---

> ### Comment · Reviewer_Mjcu · 2023-11-22
>
> Thank you for all of those responses! The Lithium-ion state tracking experiments and the worst-case disturbance settings are nice additions. Everything I originally brought up has been very clearly addressed. I just re-read the paper and the other reviews/responses in this. I have no outstanding concerns on this paper and think it will make for a nice TMLR contribution!

---

### Review · Reviewer_FBmt · 2023-09-09

**Summary Of Contributions:**

This work relaxes the assumption that the dynamical system is known which was made in the work [Yaghmaie & Modares (2023)] by using the "exploit-then-commit" type algorithm combining the algorithms of [Hazen et al (2020)] and [Yaghmaie & Modares (2023)].

**Audience:**

Yes

**Claims And Evidence:**

Yes

**Requested Changes:**

I will list some major/minor concerns here.

Major:
1. Definition 1 is a bit vague; what do you mean by “This can also be represented as…”.  Is it a part of the definition or some derived properties?
2. In Remark 1 in the paper [Yaghmaie & Modares (2023)]; it is mentioned that it is a trivial extension to the case where the system is unknown.  And indeed this seems the trivial combinations of the algorithm 2 of [Hazen et al (2020)] and the mentioned work.
3. The stabilizing controller gain K is known?  And K in Algorithm 2 seems strangely typed.
4. Figures and the legends are strange.  Please refine the experimental sections carefully.
5. Please describe briefly what the technical novelty for proving the bound in the main body.

Minor:
1. page 1 after “an instantaneous cost function” maybe better to wrap references; and others
2. page 1 Despite this advantage, existing RL-based control…  is it for continuous state/control case?
3. page 2 “the presented online learning…” the must be The.
4. page 2 “Estimating this expected cost, however, is only computationally tractable if the noise is Gaussian” isn’t it too strong statement?
5. page 2 “sqrt(O(T)) → O(sqrt(T))”?
6. Related work: may need “:” after paragraph
7. LQR controller may provide some guarantee but it hasn’t yet provided it for non-Gaussian cases.
8. Definition 2: G_lambda is full-row rank, and … → G_lambda defined by … where … is full-row rank?  … is also → is said to be… ?  Perhaps move the sentence “controllability index…” out of the definition.
9. What is “identified dynamical system”?  Better to rephrase it or define it properly.
10. Define Lhat.
11. Memory augmented policy has been used in the literature; please cite some of them before presenting the definition.
12. what do you mean by “general convex function” in page 6
13. page 6 what is “:” between truncated control and cost f?
14. L2 norm it is mathecal in some part and not in some part.

**Strengths And Weaknesses:**

Strength:
1. This paper analyzes the regret bound thoroughly for the presented algorithm
2. This paper compares the presented algorithm against the typical control algorithms.

Weakness:
1. It seems that the work is a trivial extension from [Yaghmaie & Modares (2023)].
In Remark 1 in the paper [Yaghmaie & Modares (2023)]; it is mentioned that it is a trivial extension to the case where the system is unknown.  And indeed this seems the trivial combination of the algorithm 2 of [Hazen et al (2020)] and [Yaghmaie & Modares (2023)].
2. The main technical novelty for proving the bound is not really summarized in the main body of the paper; making it hard to see the novelty beyond the trivial combination of the above.
3. Figures and the legends are strange in experimental section.
4. It is not really well explained how the initial stabilizing controller gain K is obtained.

---

> ### Author Response · Authors · 2023-11-21
>
> Thank you for your positive feedback on our manuscript. We have revised the entire paper, addressing identified issues and incorporating feedback from the reviewers, all of which are highlighted in blue throughout the document.  We have incorporated a real-world experiment centered on tracking constant states of the Lithium-ion battery circuit, which is now featured in the Appendix for a comprehensive analysis of the algorithm. We hope that the current manuscript aligns with the standards of TMLR. Your time and valuable insights into our paper are greatly appreciated.  Please see our point-to-point response to your comments.
>
> {Major}:
>
> 1. You are right. We have now modified Definition 1 to avoid ambiguity. The new definition now matches the standard definition of $(\kappa,\gamma)$-stable (Agarwal et al., 2019).
>
> 2. We believe that it is not trivial to extend the results of [Yaghmaie \& Modares (2023)] to systems with unknown dynamics in practice, even though conceptually, it might seem trivial as it combines system identification with the memory-augmented control. Integrating system identification into the memory-augment control algorithm presents challenges in regret analysis, which requires new analysis and learning rates. This regret analysis provides great insight into the control of unknown systems.  Besides, another aspect is to balance the right balance between the identification and control horizons to guarantee a sublinear regret bound.
>
> 3. Yes, the stabilizing controller that can make the unknown system $(\kappa,\gamma)-\text{stable}$ is assumed to be known in the newly added Assumption 4 of the paper.  We have also added Remark 1 to explain how this set of known initial control gains may be attained.
>
> 4. We have now revised the experiment section and the graphs specifically. Also, we have now added Assumption 4, Remark 1, and also in the statement of the experiment design on page 14 on how one can obtain the initial stabilizing control gain. In addition to that, we also provided another example in the Appendix section to provide more empirical results on the control design performance.
>
> 5. We have now brought the proof of our regret analysis in the body of the paper. The main technical difficulty in the regret analysis lies in combining the system identification and the online control approach where both the estimated and true dynamics are present. Another aspect is to balance the right balance between the identification and control horizons to guarantee a sublinear regret bound.
>
> {Minor:}
>
> 1. Thank you! We have now fixed that.
> 2. Thank you for mentioning this. We meant problems and solutions related to continuous state and action space in the introduction. We have now added that to our argument.
> 3. We have now fixed that.
> 4. We have now changed the statement to avoid misunderstanding.
> 5. We have now fixed it.
> 6. Thank you. We have now added ":" where it was missing in the Related works.
> 7. You are right, we changed our statement in the Related work part.
> 8. Thank you. We now moved the explanation of the controllable system out of the definition.
> 9. By identified system we meant the system whose matrices have been estimated using the identification algorithm. We now have fixed the statement to make it more clear.
> 10. We defined $\hat{L}$ and also modified the statement of the Lemma to make it clearer.
> 11. Thank you! We cited the paper that we were inspired by to develop this algorithm.
> 12. We have revised our statement to clarify that the cost function in our paper is not strictly limited to quadratic forms; rather, it can encompass a wider array of cost functions, as long as they adhere to the assumption of convexity.
> 13. We have now fixed that.
> 14. Thank you for pointing that out. We have now fixed that issue.

---

> > ### Comment · Reviewer_FBmt · 2023-11-22
> > **Thank you for the response**
> >
> > Thank you for the responses; my concerns have been addressed thoroughly.
> > Based on the solidity of this work, I think this paper meets the standard of TMLR.

---

### Review · Reviewer_X7qZ · 2023-11-13

**Summary Of Contributions:**

This paper presents a novel approach for designing reference tracking controllers for linear systems under uncertainties and adversarial disturbances.  The paper tackles the challenges not addressed by prior works, including unknown system and exosystem dynamics, and adversarial disturbances that are beyond bounds or probability distributions. The paper also introduces a (disturbance, reference)-action control policy that leverages past disturbances and reference values, enabling adaptation to adversarial settings.The proposed algorithm first estimates unknown dynamics and uncertainties, then learns a parameterized memory-augmented controller, optimizing a convex cost function while accounting for identification errors, unknown exosystem dynamics, and adversarial disturbances. The paper proves a regret bound of $O(T^\frac{2}{3}))$ for the proposed algorithm, demonstrating its effectiveness compared to the $O(\sqrt{T})$ bound achieved by previous approaches with known system dynamics. The authors show simulation results that compare the proposed controller to $H_{\infty}$ and LQR control, demonstrating its superior performance in dealing with uncertainties and adversarial conditions. This work offers contributions to the field of control theory by enabling robust tracking control under challenging real-world uncertainties and adversarial influences.

**Audience:**

Yes

**Broader Impact Concerns:**

No concerns.

**Claims And Evidence:**

Yes

**Requested Changes:**

Please provide empirical evidence in more complex and realistic environment and tasks.

**Strengths And Weaknesses:**

Strengths:

1. The proposed approach is novel and of value to the control community.

2. The proposed method is theoretically sound. The authors presented thorough analysis of the approach and showed that the method can achieve slightly worse regret bound but with unknown dynamics, which is a great result.

3. The authors performed simulated experiments that show the method can outperform prior methods such as $H_{\infty}$ and LQR control methods in linear control.

Weaknesses:

1. The experiments done in the paper are fairly simple and done in toy environments. It is unclear if the proposed method can work well in more complex and realistic environment. The method is motivated by working well in uncertainties and disturbance. Therefore, it is even more important to show some experiments in realistic tasks that can demonstrate the effectiveness and practicality of the proposed method.

---

> ### Author Response · Authors · 2023-11-21
>
> Thank you for your positive assessment of our proposed approach. We're delighted to hear that you find our method novel and valuable to the control community. Your recognition of the theoretical soundness, thorough analysis, and favorable performance demonstrated in simulated experiments is truly encouraging. If you have any further comments or suggestions, we would be grateful to hear them.
>
> To answer your concern, because of the limitations of C.E. $LQR$ and C.E. $H_{\infty}$ state tracking algorithms, it is challenging to find a system in which all the states are tracked with fewer control inputs than the states of the system. This is due to the fact that when computing the feedforward gain, we will need for these algorithms that the input matrix $B$ be invertible. It should be mentioned that this is not a limitation of our presented control algorithm and actually, it is one of the advantages of our method over those. That said, we included a new real-world experiment focusing on tracking Lithium-ion battery constant states. This has been added to the Appendix to present a complete analysis and assessment of the algorithm. We have also revised the whole paper and addressed issues that we found plus the ones mentioned kindly by the reviewers. All the revisions are highlighted in blue color throughout the paper.  We hope the paper in the current format matches the standards of TMLR. We really appreciate your time and input on our paper.

---

### Decision · Action_Editor_5TNG · 2023-12-27

**Recommendation:** Accept as is

**Comment:**

All reviewers agree, and I concur, that the paper contributes an important and thoroughly analyzed and evaluated method for trajectory tracking of a linear system under disturbances. The contributed algorithm is a collection of the dispersed previous ideas -- nevertheless the paper offer a solid contribution of a not well-explored problem. The authors successfully addressed all the reviewers questions.

Optionally, I would recommend that the authors look into [1] as an example for a work that addresses online compensation for unknown stochastic disturbances on control affine systems with the unknown dynamics, for a goal-conditioned navigation.

[1] A. Faust, N. Malone and L. Tapia, "Preference-balancing motion planning under stochastic disturbances," 2015 IEEE International Conference on Robotics and Automation (ICRA), Seattle, WA, USA, 2015, pp. 3555-3562, doi: 10.1109/ICRA.2015.7139692.

**Audience:**

This is paper is of interest to the learning-based control theory and robotics communities.

**Claims And Evidence:**

The paper contributes state tracking method for unknown linear systems under unknown disturbances via a two-stage algorithm. The first stage of the algorithm identifies the system, while second stage learns to compensate for the control errors dynamics. The paper offers a detailed theoretical analysis and comparisons with the baselines.

---

> ### Author Response · Authors · 2024-01-08
> **Thank you!**
>
> Dear Dr. Faust,
>
> We would like to express our sincere gratitude for accepting our paper for publication in TMLR. In response to your comment, we have integrated the comparison into our paper, as suggested. We have now added the camera-ready version of the paper.
>
> We would like to thank the reviewers as well. Their thorough reading and insightful comments have been invaluable in enhancing the quality of our work. We truly appreciate your constructive feedback!
>
> Sincerely,
>
> on behalf of all authors